# Neural Collapse versus Low-rank Bias: Is Deep Neural Collapse Really Optimal?

**Peter Súkeník**
Institute of Science and Technology Austria
3400 Klosterneuburg, Austria
`peter.sukenik@ista.ac.at`

**Christoph Lampert**[*]
Institute of Science and Technology Austria
3400 Klosterneuburg, Austria
`chl@ista.ac.at`

**Marco Mondelli**[*]
Institute of Science and Technology Austria
3400 Klosterneuburg, Austria
`marco.mondelli@ista.ac.at`

## Abstract

Deep neural networks (DNNs) exhibit a surprising structure in their final layer known as neural collapse (NC), and a growing body of works has currently investigated the propagation of neural collapse to earlier layers of DNNs – a phenomenon called deep neural collapse (DNC). However, existing theoretical results are restricted to special cases: linear models, only two layers or binary classification. In contrast, we focus on non-linear models of arbitrary depth in multi-class classification and reveal a surprising qualitative shift. As soon as we go beyond two layers or two classes, DNC stops being optimal for the deep unconstrained features model (DUFM) – the standard theoretical framework for the analysis of collapse. The main culprit is a low-rank bias of multi-layer regularization schemes: this bias leads to optimal solutions of even lower rank than the neural collapse. We support our theoretical findings with experiments on both DUFM and real data, which show the emergence of the low-rank structure in the solution found by gradient descent.

## 1 Introduction

What is the geometric structure of layers and learned representations in deep neural networks (DNNs)? To address this question, Papyan et al. [39] focused on the very last layer of DNNs at convergence and experimentally measured what is now widely known as Neural Collapse (NC). This phenomenon refers to four properties that simultaneously emerge during the terminal phase of training: feature vectors of training samples from the same class collapse to the common class-mean (NC1); the class means form a simplex equiangular tight frame or an orthogonal frame (NC2); the class means are aligned with the rows of the last layer's weight matrix (NC3); and, finally, the classifier in the last layer is a nearest class center classifier (NC4). Since the influential paper [39], a line of research has aimed at explaining the emergence of NC theoretically, mostly focusing on the unconstrained features model (UFM) [36]. In this model, motivated by the network's perfect expressivity, one treats the last layer's feature vectors as a free variable and explicitly optimizes them together with the last layer's weight matrix, "peeling off" the rest of the network [9, 23]. With UFM, the NC was demonstrated in a variety of settings, both as the global optimum and as the convergence point of gradient flow.

The emergence of the NC in the last layer led to a natural research question – does some form of collapse propagate beyond the last layer to earlier layers of DNNs? A number of empirical works

---

[*]Equal contribution

38th Conference on Neural Information Processing Systems (NeurIPS 2024).

[20, 17, 43, 40, 35] gave evidence that this is indeed the case, and we will refer to this phenomenon as Deep Neural Collapse (DNC). On the theoretical side, the optimality of the DNC was obtained *(i)* for the UFM with two layers connected by a non-linearity in [51], *(ii)* for the UFM with several linear layers in [7], and *(iii)* for the deep UFM (DUFM) with non-linear activations in the context of binary classification [48]. No existing work handles the general case in which there are *multiple classes* and the UFM is *deep* and *non-linear*.

In this work, *we close the gap and reveal a surprising behavior* not occurring in the simpler settings above: for multiple classes and layers, the DNC as formulated in previous works is *not an optimal solution* of DUFM. In particular, the class means at the optimum do not form an orthogonal frame (nor an equiangular tight frame), thus violating the second property of DNC.

Let $L$ and $K$ denote the number of layers and classes, respectively. Then, if either $L \geq 3$ and $K \geq 10$ or $L \geq 4$ and $K \geq 6$, we provide an explicit combinatorial construction of a class of solutions that outperforms DNC. Specifically, the loss achieved by our construction is a factor $K^{(L-3)/(2L+2)}$ lower than the loss of the DNC solution. Our result holds as long as all matrices are regularized.

We also identify the reason behind the sub-optimality of DNC: a *low-rank bias*. Intuitively, this bias arises from the representation cost of a DNN with $l_2$ regularization, which equals the Schatten-$p$ quasi norm [37] in the deep linear case. The quasi norm is well approximated by the rank, and this intuition carries over to the non-linear case as well. In fact, the rank of our construction is $\Theta(\sqrt{K})$, while the rank of the DNC solution is $K$. We note that after the application of the ReLU, the rank of the final layer is again equal to $K$, in order to fit the training data. We also show that the first property of neural collapse (convergence to class means) continues to be strictly optimal even in this general setting and its deep counterpart is approximately optimal with smoothed ReLU activations.

We support our theoretical results with empirical findings in three regimes: *(i)* DUFM training, *(ii)* training on standard datasets (MNIST [31], CIFAR-10 [28]) with DUFM-like regularization, and *(iii)* training on standard datasets with standard regularization. In all cases, gradient descent retrieves solutions with very low rank, which can exhibit symmetric structures in agreement with our combinatorial construction, see e.g. the lower-right plot of Figure 4. We also investigate the effect of three common hyperparameters – weight decay, learning rate and width – on the rank of the solution at convergence. On the one hand, high weight decay, high learning rate and small width lead to a strong low-rank bias. On the other hand, small (yet still non-zero!) weight decay, small learning rate or large width (and more complex datasets as well) lead to a higher-rank solution, even if that is not the global optimum, and this solution often coincides with DNC, which is in agreement with earlier experimental evidence. Altogether, our findings show that if a DNC solution is found, it is not because of its global optimality, but just because of an implicit bias of the optimization procedure.

The implications of our results go beyond *deep* neural collapse. In fact, our theory suggests that even the NC2 in the last layer is not optimal, and this is corroborated by our experiments, where the singular value structure of the last layer's class-mean matrices is imbalanced, ruling out orthogonality. This means that standard single-layer UFM, as well as its deep-linear or two-layer extensions, are not sufficient to describe the full picture, as they display a qualitatively different phenomenology.

## 2  Related work

**Neural Collapse.**   Several papers (a non-exhaustive list includes [12, 15, 5, 32, 33, 58]) use neural collapse as a practical tool in applications, among which OOD detection and transfer learning are the most prevalent. On the theoretical side, the emergence of NC has been investigated, with the majority of works considering some form of UFM [36, 9]. [55, 34] show global optimality of NC under the cross-entropy (CE) loss, and [59] under the MSE loss. Similar results are obtained by [9, 49, 18, 8] for the class-imbalanced setting. [61, 23, 59] refine the analysis by showing that the loss landscape of the UFM model is benign – all stationary points are either local minima or strict saddle points which can be escaped by conventional optimizers. A more loss-agnostic approach connecting CE and MSE loss is considered in [60]. NC has also been analyzed for a large number of classes [25], in an NTK regime [45], or in graph neural networks [27]. We refer the reader to [26] for a survey.

The emergence of NC has also been studied through the lens of the gradient flow dynamics. [36] considers MSE loss and small initialization, and [16] a renormalized gradient flow of the last layer's features after fixing the last layer's weights to be conditionally optimal. [23] studies the CE loss dynamics and shows convergence in direction of the gradient flow to a KKT point of the max-margin

problem of the UFM, extending a similar analysis for the last layer's weights in [47]. The convergence speed under both losses is described in [53]. Going beyond UFM, [57, 41, 42, 29] study the emergence of NC in homogeneous networks under gradient flow; [38] provides sufficient conditions for neural collapse; and [52] perturbs the unconstrained features to account for the limitations of the model.

More recently, [20] mentions a possible propagation of the NC to earlier layers of DNNs, giving preliminary measurements. These are then significantly extended in [17, 43, 11, 40], which measure the emergence of some form of DNC in DNNs. On the theoretical front, an extension to a two-layer non-linear model is provided in [51], to a deep linear model in [7, 14] and to a deep non-linear model for binary classification in [48]. Alternatively to DUFM, [4] studies DNC in an end-to-end setting with a special layer-wise training procedure.

**Low-rank bias.** The low-rank bias is a well-known phenomenon, especially in the context of matrix/tensor factorization and deep linear networks (see e.g. [2, 6, 44, 24, 54]). For non-linear DNNs, [50] studies the gradient flow optimization of ReLU networks, giving lower and upper bounds on the average soft rank. [13] studies SGD training on deep ReLU networks, showing upper bounds on the rank of the weight matrices as a function of batch size, weight decay and learning rate. [30] proves several training invariances that may lead to low-rank, but the results require the norm of at least one weight matrix to diverge and the architecture to end with a couple of linear layers. [3] presents bounds on the singular values of non-linear layers in a rather generic setting, not necessarily at convergence. More closely related to our work is [37], which considers a deep linear network followed by a single non-linearity and then by a single layer. Their arguments to study the low-rank bias are similar to the intuitive explanation of Section 4. [19] shows that increasing the depth results in lower effective rank of the penultimate layer's Gram matrix both at initialization and at convergence. The true rank is also measured, but on rather shallow networks and it is far above the DNC rank. [1] shows a strong low-rank bias of sharpness-aware minimization, although only in layers where DNC does not yet occur and the rank is high. [10, 22] study special functional ranks (Jacobi and bottleneck) of DNNs, providing asymptotic results and empirical measurements. These results are refined in [21, 56], which show a bottleneck structure of the rank both experimentally and theoretically. The measurements of the singular values at convergence in [56] are in agreement with those of Section 6.3. We highlight that *none* of the results above allows to reason about DNC optimality, as they focus on infinite width/depth, effective or functional ranks, orthogonal settings, or are not quantitative enough.

## 3 Preliminaries

We study the class balanced setting with $N = Kn$ samples from $K$ classes, $n$ per class. Let $f(x) = W_L \sigma(W_{L-1} \sigma(\dots W_1 \mathcal{B}(x) \dots))$ be a DNN with backbone $\mathcal{B}(\cdot)$. The backbone represents the majority of the deep network *before* the last $L$ layers, e.g. the convolutional part of a ResNet20. Let $X \in \mathbb{R}^{d \times N}$ be the training data, and $H_1 = \mathcal{B}(X) \in \mathbb{R}^{d_1 \times N}, H_2 = \sigma(W_1 H_1) \in \mathbb{R}^{d_2 \times N}, \dots, H_L = \sigma(W_{L-1} H_{L-1}) \in \mathbb{R}^{d_L \times N}$ its feature vector representations in the last $L$ layers, with $\tilde{H}_l$ denoting their counterparts before applying the ReLU $\sigma$. We refer to $h_{ci}^l$ and $\tilde{h}_{ci}^l$ as to the $i$-th sample of $c$-th class of $H_l$ and $\tilde{H}_l$, respectively. Let $\mu_c^l = \frac{1}{n} \sum_{i=1}^n h_{ci}^l$ and $\tilde{\mu}_c^l = \frac{1}{n} \sum_{i=1}^n \tilde{h}_{ci}^l$ be the class means at layer $l$ after and before applying $\sigma$, and $M_l, \tilde{M}_l$ the matrices of the respective class means stacked into columns. We organize the training samples so that the labels $Y \in \mathbb{R}^{K \times N}$ equal $I_K \otimes \mathbf{1}_n^T$, where $I_K$ is a $K \times K$ identity matrix, $\otimes$ is the Kronecker product and $\mathbf{1}_n$ the all-one vector of size $n$.

**Deep neural collapse (DNC).** As there are no biases in our network model, the second property of DNC requires the class mean matrices to be orthogonal (instead of forming an ETF) [43, 48].

**Definition 1.** *We say that layer $l$ exhibits DNC 1, 2 or 3 if the corresponding conditions are satisfied (the properties can be stated for both after and before the application of ReLU):*

DNC1: *The within-class variability of either $H_l$ or $\tilde{H}_l$ is 0. Formally, $h_{ci}^l = h_{cj}^l$, $\tilde{h}_{ci}^l = \tilde{h}_{cj}^l$ for all $i, j \in [n]$ or, in matrix notation, $H_l = M_l \otimes \mathbf{1}_n^T$, $\tilde{H}_l = \tilde{M}_l \otimes \mathbf{1}_n^T$.*

DNC2: *The class-mean matrices $M_l, \tilde{M}_l$ are orthogonal, i.e., $M_l^T M_l \propto I_K$, $\tilde{M}_l^T \tilde{M}_l \propto I_K$.*

DNC3: *The rows of the weight matrix $W_l$ are either 0 or collinear with one of the columns of the class-means matrix $M_l$.*

**Deep unconstrained features model.** To define DUFM, we generalize the model in [48] to an arbitrary number of classes $K$.

**Definition 2.** *The $L$-layer deep unconstrained features model ($L$-DUFM) denotes the following optimization problem:*

$$\min_{H_1, W_1, \ldots, W_L} \frac{1}{2N} \|W_L \sigma(W_{L-1}\sigma(\ldots W_2\sigma(W_1 H_1)\ldots)) - Y\|_F^2 + \sum_{l=1}^{L} \frac{\lambda_{W_l}}{2} \|W_l\|_F^2 + \frac{\lambda_{H_1}}{2} \|H_1\|_F^2,$$
$$(1)$$

*where $\|\cdot\|_F$ denotes the Frobenius norm and $\lambda_{H_1}, \lambda_{W_1}, \ldots, \lambda_{W_L} > 0$ are regularization parameters.*

## 4 Low-rank solutions outperform deep neural collapse

**Intuitive explanation of the low-rank bias.** Consider a simplified version of $L-$DUFM:

$$\min_{H_1, W_1, \ldots, W_L} \frac{1}{2N} \|W_L \sigma(W_{L-1}\ldots W_2 W_1 H_1) - Y\|_F^2 + \sum_{l=1}^{L} \frac{\lambda_{W_l}}{2} \|W_l\|_F^2 + \frac{\lambda_{H_1}}{2} \|H_1\|_F^2. \quad (2)$$

Compared to (1), (2) removes all non-linearities except in the last layer, making the remaining part of the network a deep linear model, a construction similar to the one in [37]. Now, we leverage the variational form of the Schatten-$p$ quasi-norm [46], which gives

$$c \left\| \tilde{H}_L \right\|_{S_{2/L}}^{2/L} = \min_{H_1, W_1, \ldots, W_{L-1}: H_1 W_1 \ldots W_{L-1} = \tilde{H}_L} \sum_{l=1}^{L-1} \frac{\lambda_{W_l}}{2} \|W_l\|_F^2 + \frac{\lambda_{H_1}}{2} \|H_1\|_F^2,$$

where $c$ can be computed explicitly. Thus, after solving for $H_1, W_1, \ldots, W_{L-1}$, the simplified $L$-DUFM problem (2) can be reduced to

$$\min_{\tilde{H}_L W_L} \frac{1}{2N} \left\| W_L \sigma(\tilde{H}_L) - Y \right\|_F^2 + \frac{\lambda_{W_L}}{2} \|W_L\|_F^2 + \frac{\lambda_{\tilde{H}_L}}{2} \left\| \tilde{H}_L \right\|_{S_{2/L}}^{2/L}.$$

For large values of $L$, $\left\| \tilde{H}_L \right\|_{S_{2/L}}^{2/L}$ is well approximated by the rank of $\tilde{H}_L$. Hence, the objective value is low when the output $W_L H_L$ fits $Y$ closely, while keeping $\tilde{H}_L$ low-rank, which justifies the low-rank bias. Crucially, the presence of additional non-linearities in the $L$-DUFM model (1) does not change this effect much, as long as one is able to define solutions for which most of the intermediate feature matrices $\tilde{H}_l$ are non-negative (so that ReLU does not have an effect).

**Low-rank solution outperforming DNC.** We define the combinatorial solution that outperforms DNC, starting from the graph structure on which the construction is based.

**Definition 3.** *A triangular graph $\mathcal{T}_n$ of order $n$ is a line graph of a complete graph $\mathcal{K}_n$ of order $n$. $\mathcal{T}_n$ has $\binom{n}{2}$ vertices, each representing an edge of the complete graph, and there is an edge between a pair of vertices if and only if the corresponding edges in the complete graph share a vertex. Moreover, let $T_n$ be the normalized incidence matrix of $\mathcal{K}_n$, i.e., $(T_n)_{i,j} = \frac{1}{\sqrt{n-1}}$ if vertex $i$ belongs to edge $j$ and 0 otherwise. Let $G_n$ denote the adjacency matrix of $\mathcal{T}_n$.*

We recall that $\mathcal{T}_n$ is a strongly regular graph with parameters $(n(n-1)/2, 2(n-2), n-2, 4)$ and spectrum $2(n-2)$ with multiplicity 1, $n-4$ with multiplicity $n-1$ and $-2$ with multiplicity $n(n-3)/2$. Next, we construct an explicit solution $(H_1, W_1, \ldots, W_L)$ based on the triangular graph. For ease of exposition, we focus on the case where the number of classes $K$ equals $\binom{r}{2}$ for some $r \geq 4$, deferring the general definition to Appendix A.1.

**Definition 4.** *Let $K = \binom{r}{2}$ for $r \geq 4$. Then, a strongly regular graph (SRG) solution of the $L$-DUFM problem (1) is obtained by setting the matrices $(H_1, W_1, \ldots, W_L)$ as follows:*

- *For all $l$, the feature matrices $H_l, \tilde{H}_l$ are DNC1 collapsed, i.e., $H_l = M_l \otimes \mathbf{1}_n^T, \tilde{H}_l = \tilde{M}_l \otimes \mathbf{1}_n^T$.*

- *For $2 \leq l \leq L-1$, $M_l = \tilde{M}_l$, each row of $\tilde{M}_l$ is a non-negative multiple of a row of $T_r$ (as in Definition 3), and the sum of squared norms of the rows of $\tilde{M}_l$ corresponding to a row of $T_r$ is the same for each row of $T_r$. Since $\tilde{M}_l$ is entry-wise non-negative, $M_l = \tilde{M}_l$.*

- *For $l = 1$, $W_1, M_1$ are any pair of matrices minimizing the objective conditionally on $M_2$ defined above.*

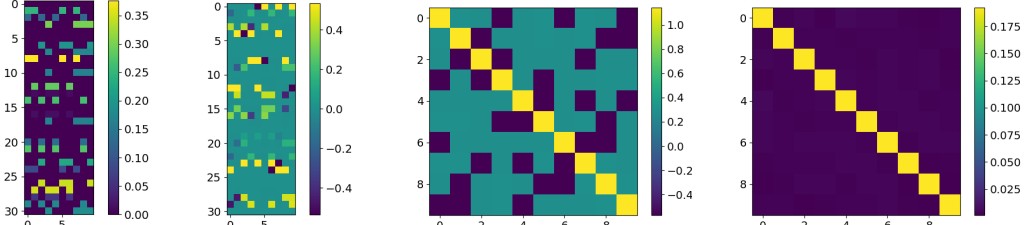

Figure 1: Strongly regular graph (SRG) solution with $L = 4$, $K = 10$ and $r = 5$. **Left:** Class-mean matrix of the third layer $M_3$. The non-zero entries of each row have the same value and their number is $r - 1$, which corresponds to the degree of the complete graph $\mathcal{K}_r$. **Middle:** Class-mean matrix of the fourth layer before ReLU $\tilde{M}_4$ (**middle left**), and its Gram matrix $\tilde{M}_4^T \tilde{M}_4$ (**middle right**). The SRG construction has very low rank before ReLU: $\text{rank}(\tilde{M}_4) = r$ and $\text{rank}(\sigma(\tilde{M}_4)) = K$. **Right:** $\tilde{M}_4^T \tilde{M}_4$ for DNC. The DNC solution has rank $K$ in all layers before and after ReLU.

- *For $l \geq 2$, $W_l$ minimizes the objective conditional to input and output to that layer.*

- *As for the last layer $L$, let $A_L$ be a $K \times r$ matrix where the set of rows equals the set of vectors with two $(-1)$ entries and $r - 2$ $(+1)$ entries. Then, $M_L = \sigma(\tilde{M}_L)$, the rows of $\tilde{M}_L$ are a non-negative multiple of $A_L T_r$, and the sum of their squared norms corresponding to either row of $A_L T_r$ is equal.*

- *Finally, the Frobenius norms (i.e. scales) of $M_1, W_1, \ldots, W_L$ are chosen so as to minimize (1) while satisfying the construction above.*

In this construction, columns and rows of class-mean matrices are associated to edges and vertices of the complete graph $\mathcal{K}_r$. Each row (corresponding to a vertex) has non-zero entries at columns that correspond to edges containing the vertex. In the final layer, each row of $\tilde{M}_L$ corresponds to a weighting of vertices in $\mathcal{K}_r$ s.t. exactly two vertices get $-1$ weight and the rest $+1$, and the value at a column is the sum of the values of the vertices of the edge. The class-mean matrices of the SRG solution are illustrated in Figure 1 for $L = 4$ and $K = 10$ (which gives $r = 5$): we display $M_3, \tilde{M}_4, \tilde{M}_4^T \tilde{M}_4$ and, for comparison, also $\tilde{M}_4^T \tilde{M}_4$ of a DNC solution. Very similar solutions to SRG are shown for $K = 6$ and $K = 15$ in Figures 7 and 8 of Appendix B.1.

Let us highlight the properties of the SRG solution, which are crucial to outperform DNC. First, the rank of the intermediate feature and weight matrices is very low, only of order $\Theta(K^{1/2})$, since by construction there are only $r = \Theta(K^{1/2})$ linearly independent rows. This is contrasted with the DNC solution that has rank $K$ in all intermediate feature and weight matrices. The low rank of the SRG solution is due to the specific structure of the triangular graph, which has many eigenvalues equal to $-2$ that become 0 after adding twice a diagonal matrix. Second, the definition of $\tilde{M}_L$ ensures that $M_L = \sigma(\tilde{M}_L)$ has full rank $K$. This allows the output $W_L M_L$ to also have full rank and, therefore, fit the identity matrix $I_K$, thus reducing the first term in the loss (1). Finally, the highly symmetric nature of the SRG solution balances the feature and weight matrices so as to minimize large entries and, therefore, the Frobenius norms, thus reducing the other terms in the loss (1).

**Main result.** For any $L$-DUFM problem (specified by $K, n$ and all the regularization parameters), let $\mathcal{L}_{SRG}, \mathcal{L}_{DNC}$ be the losses incurred by the SRG and DNC solutions, see Definitions 4 and 1, respectively. At this point we are ready to state our key result.

**Theorem 5.** *If $K \geq 6, L \geq 4$ or $K \geq 10, L = 3$ and $d_l \geq K$ for all $l$, then $\mathcal{L}_{SRG} < \mathcal{L}_{DNC}$. Moreover, consider any sequence of $L$-DUFM problems for which $K \to \infty$ so that $0.499 > \mathcal{L}_{DNC}$ for each problem. In that case,*

$$\frac{\mathcal{L}_{SRG}}{\mathcal{L}_{DNC}} = \mathcal{O}(K^{\frac{3-L}{2(L+1)}}). \tag{3}$$

In words, as long as the number of classes and layers is not too small, the SRG solution always outperforms the collapsed one and the gap grows with the number of classes $K$.

The proof first computes the conditionally optimal values of $\|W_l\|_F^2$ for both the SRG and DNC solutions. The specific structure of these solutions enables to calculate pseudoinverses of the intermediate features, thus enabling the explicit computation of the weight norms. All these values depend only on the singular values of the feature matrices, which are explicitly given by their scale. As a result, both $\mathcal{L}_{SRG}$ and $\mathcal{L}_{DNC}$ are expressed via an optimization problem in a single scalar variable and, by comparing these problems, the statement follows. The details are deferred to Appendix A.1.

Although the argument requires $L = 3, K \geq 10$ or $L \geq 4, K \geq 6$, the experiments in Appendix B.1 show that the DNC solution is not optimal when $L \geq 4, K \geq 3$ or $L = 3, K \geq 7$. Furthermore, for $L = 3$ and large $K$, there is a large gap between $\mathcal{L}_{SRG}$ and $\mathcal{L}_{DNC}$ (even if (3) trivializes). For either $K = 2$ or $L = 2$, the DNC is optimal, as shown in [51, 48].

## 5 Within-class variability collapse is still optimal

While the DNC2 property conflicts with the low-rank bias, the same is not true for DNC1, as the within-class variability collapse supports a low rank. We show below that the last-layer NC1 property remains optimal for any $L$-DUFM problem. A proof sketch follows, with the complete argument deferred to Appendix A.2.

**Theorem 6.** *The optimal solutions of the $L$-DUFM* (1) *exhibit DNC1 at layer L, i.e.,*

$$H_L^* = M_L^* \otimes \mathbf{1}_n^T$$

*holds for any optimal solution $(H_1^*, W_1^*, \ldots, W_L^*)$ of the $L$-DUFM problem.*

*Proof sketch:* Assume by contradiction that there exists an optimal solution of (1) with regularization parameters $(\lambda_{H_1}, \lambda_{W_1}, \ldots, \lambda_{W_L})$, denoted as $(H_1^*, W_1^*, \ldots, W_L^*)$, which does not exhibit neural collapse at layer $L$. Then, we can construct two *different* optimal solutions of the $L$-DUFM problem with $n = 1$ and regularization parameters $(n\lambda_{H_1}, \lambda_{W_1}, \ldots, \lambda_{W_L})$ of the form $(H_1^{(1)}, W_1^*, \ldots, W_L^*)$ and $(H_1^{(2)}, W_1^*, \ldots, W_L^*)$. These two solutions share the weight matrices, and $H_1$ (and, therefore, $H_L$) only differs in a single column (w.l.o.g., the first column). The optimality of these solutions can be proved using separability and symmetry of the loss function w.r.t. the columns of $H_1$.

Denote the first (differing) columns of $H_L^{(1)}$ and $H_L^{(2)}$ as $x$ and $y$, respectively. By exploiting the linearity of the loss function on a ray $\{th_{11}^1, t \geq 0\}$ for any $h_{11}^1$, a direct computation gives that $x$ and $y$ are not aligned. Let $\mathcal{L}$ be the loss in (1). By optimality of both solutions, we get

$$\left.\frac{\partial\mathcal{L}}{\partial W_L}\right|_{(H_1, W_1, \ldots, W_L)=(H_1^{(1)}, W_1^*, \ldots, W_L^*)} = 0 = \left.\frac{\partial\mathcal{L}}{\partial W_L}\right|_{(H_1, W_1, \ldots, W_L)=(H_1^{(2)}, W_1^*, \ldots, W_L^*)}. \quad (4)$$

An application of the chain rule gives

$$\frac{\partial\mathcal{L}}{\partial W_L} = \frac{\partial\mathcal{L}_F}{\partial\tilde{H}_{L+1}}\frac{\partial\tilde{H}_{L+1}}{\partial W_L} + \lambda_{W_L}W_L = \frac{\partial\mathcal{L}_F}{\partial\tilde{H}_{L+1}}H_L^T + \lambda_{W_L}W_L,$$

where $\tilde{H}_{L+1}$ is the model output and $\mathcal{L}_F$ the first term of $\mathcal{L}$, corresponding to the label fit. Plugging this back into (4) and using that $W_L^*$ is the same in both expressions, we get $A(H_L^{(1)})^T = B(H_L^{(2)})^T$, where we have denoted by $A$ and $B$ the partial derivatives $\frac{\partial\mathcal{L}_F}{\partial\tilde{H}_{L+1}}$ evaluated at $(H_1^{(1)}, W_1^*, \ldots, W_L^*)$ and $(H_1^{(2)}, W_1^*, \ldots, W_L^*)$, respectively. As $\mathcal{L}_F$ is separable with respect to the columns of $H_l, \tilde{H}_l$ for all $l$, the matrices $A, B$ can only differ in their first columns (denoted by $a, b$), and they are identical otherwise. This implies that $ax^T = by^T$. After some simple considerations and using that $x$ and $y$ are not aligned, we reach a contradiction, as we conclude that $x \neq y$ is impossible. □

The difficulty in extending Theorem 6 to a result on the unique optimality of DNC1 for all layers stems from the special role of $W_L$ as the loss is differentiable w.r.t. it. By considering a differentiable relaxation of ReLU, we show below an approximate result for a *relaxed $L$-DUFM model*.

**Definition 7.** *We denote by $ReLU_\epsilon$ (or $\sigma_\epsilon$) a function satisfying the following conditions: (i) $\sigma_\epsilon(x) = \sigma(x)$, for $x \in (-\infty, 0] \cup [\epsilon, \infty)$, (ii) $0 < \sigma_\epsilon(x) < \sigma(x)$ for $x \in (0, \epsilon)$, and (iii) $\sigma_\epsilon$ is continuously differentiable with derivative bounded by a universal constant and strictly positive on $(0, \epsilon)$.*

**Theorem 8.** *Denote by L-DUFM$_\epsilon$ the equivalent of* (1)*, with $\sigma$ replaced by $\sigma_\epsilon$. Let $D = \max\{d_2, d_3, \ldots, d_L\}$ and $\bar{\lambda} = \lambda_{H_1} \lambda_{W_1} \ldots \lambda_{W_L}$, with the regularization parameters upper bounded by $1/(L+1)$. Then, for any globally optimal solution of the L-DUFM$_\epsilon$ problem, the distance between any two feature vectors of the same class in any layer is at most*

$$\frac{6\epsilon\sqrt{D(L+1)}}{(L+1)^{L+1}\bar{\lambda}\sqrt{n}}. \tag{5}$$

In words, as the activation function approaches ReLU (i.e., $\epsilon \to 0$), the within-class variability tends to 0. The proof starts with a similar strategy as the argument of Theorem 6 and then explicitly tracks the error due to replacing $\sigma$ with $\sigma_\epsilon$ through the layers. The full argument is in Appendix A.2.

## 6 Numerical results

We employ the standard DNC1 metric $\text{tr}(\Sigma_W)/\text{tr}(\Sigma_B)$, where $\Sigma_W, \Sigma_B$ are the within and between class variabilities. This is widely used in the literature [52, 43, 4] and considered more stable than other metrics [43]. We measure the DNC2 metric as the condition number of $M_l$ for $l \geq 1$ [48]. We do not measure DNC3 here, as it is not well-defined for solutions that do not satisfy DNC2. For end-to-end DNN experiments, we employ a model from [48] where an MLP with a few layers is attached to a ResNet20 backbone. The output of the backbone is then treated as unconstrained features, and DNC metrics are measured for the MLP layers.

### 6.1 DUFM training

We start with the $L$-DUFM model (1), training both features and weights. In the top row of Figure 2, we consider a 4-DUFM, with $K = 10$ and $n = 50$, presenting the training progression of the losses (left plot), the DNC1 metrics (center plot) and the singular values at convergence (right plot).

The results are in excellent agreement with our theory. First, the training loss outperforms that of the DNC solution, and it is rather close to that of the SRG solution. Second, DNC1 holds in a clear way in all layers, especially in the last ones. Third, the solution at convergence exhibits a strong low rank bias: the ranks of intermediate layers range from 5 to 8, and they are always the same in all intermediate layers within one run. For comparison, we recall that the intermediate layers of the DNC solution have full rank $K = 10$. Third, for a few runs, the Gram matrices of the intermediate class means resulting from gradient descent training coincide with those of an SRG solution. Finally we highlight that, similarly to our theory, the solutions found in all our experiments in the entire Section 6 have non-negative pre-activations in all intermediate layers of the MLP head except the last one.

**Impact of number of classes and depth.** For $K = 2$ or $L = 2$, we recover the results of [48, 51] irrespective of other hyperparameters. The higher the number of classes, the more prevalent are low-rank solutions, while finding DNC solutions becomes challenging. The same holds for increasing the number of layers. For $L = 3$ and low number of classes ($K \leq 6$), we weren't able to experimentally find solutions that would outperform DNC, which aligns nicely with the fact that SRG outperforms DNC only from $K = 10$ for $L = 3$. For large number of classes, the difference between the loss of low-rank solutions and the DNC loss is considerable already for $L = 3$ and becomes even larger for higher $L$. This is illustrated in the left plot of Figure 3.

For $L \leq 5$ and moderate number of classes ($K \leq 30$), gradient descent solutions are as follows: until layer $L - 1$, feature matrices share the same rank and have similar Gram matrices; intermediate activations are typically non-negative, and the ReLU has no effect; then, the rank jumps to $K$ after the final ReLU, as pre-activations are also negative. For large $L$ or large $K$, the rank of the first few layers is low, growing gradually in the last couple of layers (see Figure 6 in Appendix B.1); the ReLU is active only in the final layers. This means that not only very low-rank solutions outperform DNC (as shown by our theory), but such solutions are routinely reached by gradient descent.

**Impact of weight decay and width.** While neither weight decay nor width influence Theorem 5 – which shows that DNC is not optimal – both quantities influence the nature of the solutions found by gradient descent. In particular, the stronger the weight decay, the lower the rank, see the middle plot in Figure 3. For very small weight decay, DNC is sometimes recovered; for very high weight decay,

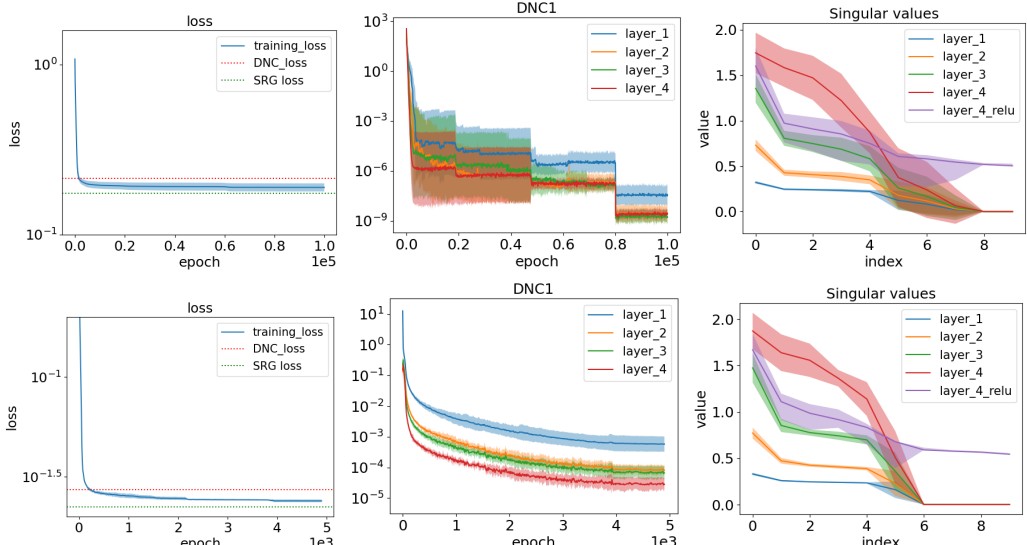

Figure 2: Training loss compared against DNC and SRG losses (**left**), DNC1 metric training progression (**middle**) and singular value distribution at convergence (**right**). **Top row:** 4-DUFM training with $K = 10$, $\lambda = 0.004$ for all regularization parameters, learning rate of $0.5$ and width $30$. Results are averaged over 10 runs, and we show the confidence intervals at 1 standard deviation. **Bottom row:** Training of a ResNet20 with a 4-layer MLP head on CIFAR10, using a DUFM-like regularization. We use weight decay $0.005$ except $\lambda_{H_1} = 0.000005$ (to compensate for $n = 5000$, which significantly influences the total regularization strength), learning rate $0.05$ and width $64$ for all the MLP layers. Results are averaged over 5 runs, and we show the confidence intervals at 1 standard deviation.

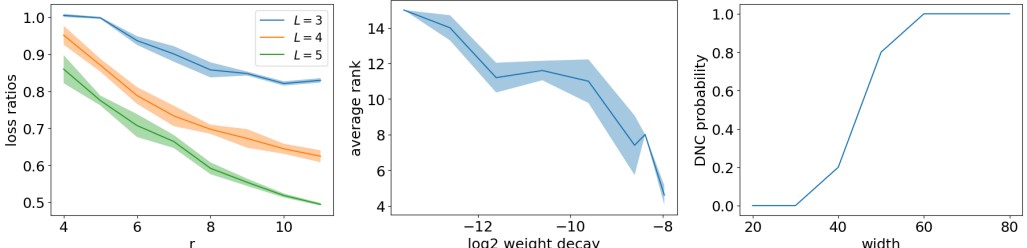

Figure 3: All experiments refer to the training of an $L$-DUFM model. Results are averaged over 5 runs, and we show the confidence intervals at 1 standard deviation. **Left:** Ratio between SRG and DNC loss ($\mathcal{L}_{SRG}/\mathcal{L}_{DNC}$), as a function of $r$, where the number of classes is $K = \binom{r}{2}$. Different curves correspond to different values of $L \in \{3, 4, 5\}$. **Middle:** Average rank at convergence, as a function of the weight decay in $\log_2$-scale, when $L = 4$ and $K = 15$. **Right:** Empirical probability of finding a DNC solution as a function of the width, when $L = 4$ and $K = 10$.

it is never recovered. The width has an opposite effect, see the right plot of Figure 3. For small width, low-rank solutions are much more likely to be found; large width has a strong implicit bias towards DNC and, thus, rank $K$ solutions. This means that, surprisingly, a larger width leads to a larger loss, since low-rank solutions exhibit a smaller loss than DNC. Thus, at least in DUFM, the infinite-width limit prevents gradient descent from finding a globally optimal solution, and sub-optimal solutions are reached with increasingly high probability.

## 6.2 End-to-end experiments with DUFM-like regularization

Next, we train a DNN backbone with an MLP head, regularizing *only* the output of the backbone and the layers of the MLP head (and not the layers of the backbone). This regularization is closer to our theory than the standard one, since we explicitly regularize the Frobenius norm of the unconstrained features. We also note that training with such a regularization scheme is easier than training with the standard regularization scheme. In the bottom row of Figure 2, we consider a ResNet20 backbone with a 4-layer MLP head trained on CIFAR10.

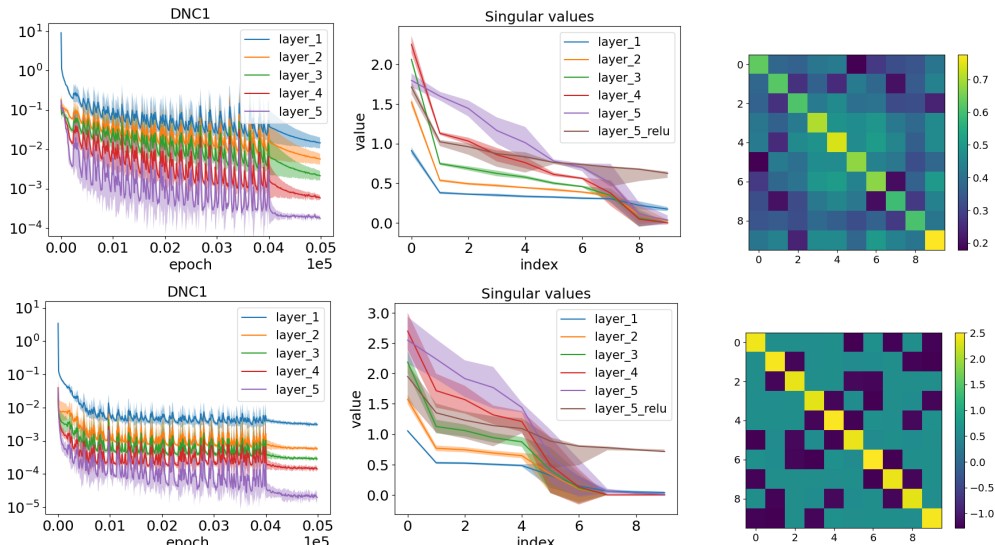

Figure 4: Training of a ResNet20 with a 5-layer MLP head on CIFAR-10 (**top row**) and MNIST (**bottom row**), using the standard regularization. We pick a large weight decay (0.08 for CIFAR-10 and 0.04 for MNIST) and a large learning rate (0.005 for CIFAR-10 and 0.01 for MNIST). Results are averaged over 5 runs, and we show the confidence intervals at 1 standard deviation. **Left:** DNC1 metric training progression. **Middle:** Singular value distributions at convergence for all the layers. **Right:** Gram matrices of $M_3$ (CIFAR-10) and $\tilde{M}_5$ (MNIST).

The results agree well with our theory, and they are qualitatively similar to those of Section 6.1 for DUFM training. The DNNs consistently outperform the DNC loss, but still achieve DNC1. The ranks of class-mean matrices range from 5 to 6, and they are always the same in all intermediate layers within one run. Remarkably, the SRG solution was found by gradient descent also in this setting.

Both weight decay and learning rate affect the average rank of the solutions found by gradient descent. Varying the width can lead to unexpected results, as it changes the ratio between the number of parameters in the MLP and that in the backbone, so the effect of the width is harder to interpret. Similar results can be seen on MNIST.

## 6.3  End-to-end experiments

Finally, we perform experiments with standard regularization and the same architecture (i.e., DNN backbone plus MLP head) as in Section 6.2. In particular, in Figure 4 we consider a ResNet20 backbone with a 5-layer MLP head trained on CIFAR10 and MNIST with standard weight regularization.

Overall, the results remain qualitatively similar to those discussed above. This demonstrates that, in spite of a different loss landscape compared to previous settings, the low-rank bias is still responsible for DNC2 not being attained. Specifically, for CIFAR10, the rank in the third layer ranges between 8 and 9, and for MNIST ranges between 5 and 7; in contrast, the DNC solution has rank $K = 10$. All DNNs display DNC1 across all layers. Remarkably, for the MNIST experiment the solution displayed in Figure 4 found by gradient descent is the SRG solution (compare the gram matrices in bottom right plot of Figure 4 with the right-most plot of Figure 1).

The difficulty of the learning task plays a significant role in this setting: when training on MNIST, it is rather easy to reach low-rank solutions and rather difficult to reach DNC solutions and the rank depends heavily on the regularization strength as shown in Figure 10 of Appendix B.3; when training on CIFAR-10, the weight decay needs to be high for the class mean matrices to be rank deficient. Moreover, the learning rate no longer exhibits a clear relation with the rank, since gradient descent diverges when the learning is too large. We also observe that the rank deficiency is the strongest in the mid-layer of the MLP head, creating a "rank bottleneck". This can be seen by a closer look at the tails of the singular values, which better match zero at intermediate layers (the green and red curves corresponding to layers 3 and 4 have tails slightly lower than the other curves). In a more precise manner, we further measured effective ranks of all the layers in Figure 4. For instance, the effective

ranks of CIFAR10 experiment layers are $8.96, 7.46, 6.88, 7.04, 7.73$), which shows the middle layer is closest to a low hard rank matrix. The rank bottleneck is also mentioned in [22, 21, 56]. In fact, these works also measure extremely low ranks, but [22, 21] do it on synthetic data with very low inner dimension, while [56] focuses on fully convolutional architectures trained with CE loss and including biases.

In summary, Figure 4 shows that both the low-rank bias and the optimality of DNC1 carry over to the standard training regime. This means that there are hyperparameter settings for which deep neural collapse, *including in the very last layer*, is not reached (and likely not even optimal). Although the sub-optimality of DNC in the last layer is not proved formally, this phenomenon is supported by evidence across all experimental settings and further corroborated by our theory where our SRG construction is far from being DNC2-collapsed in the last layer.

## 7 Conclusion

In this work, we reveal that the deep neural collapse is *not* an optimal solution of the deep unconstrained features model – the extension of the widely used unconstrained features model. This finding considerably changes our overall understanding of DNC, as all the previous models in simplified settings showed the global optimality of neural collapse and of its deep counterpart. The main culprit – the low-rank bias – makes the orthogonal frame property of DNC, and thus DNC as a whole, too high rank to be optimal. We demonstrate this low-rank bias across a variety of experimental settings, from DUFM training to end-to-end training with the standard weight regularization. While the structure of the Gram matrices of class means is not captured by orthogonal matrices (or by the ETF), the within-class variability collapse remains optimal. Our theoretical analysis proves this for the DUFM problem, and our numerical results showcase the phenomenon across various settings.

Our analysis focuses on the MSE loss, but we expect similar results to hold for the cross-entropy loss and, in particular, that the same SRG construction proposed here would still refute the optimality of DNC. We leave as an open question whether DNC1 is strictly optimal across *all* layers. While proving this would likely require new ideas, we note that *none* of our experiments converged to a solution that would not be DNC1-collapsed.

## Acknowledgments and Disclosure of Funding

Marco Mondelli is partially supported by the 2019 Lopez-Loreta prize. This research was supported by the Scientific Service Units (SSU) of ISTA through resources provided by Scientific Computing (SciComp).

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

# A  Proofs

## A.1  Low-rank solutions outperform deep neural collapse

We start by providing more detailed definitions of SRG and DNC solutions.

**Definition 9.** *Let $K = \binom{r}{2}$ for $r \geq 4$. Then, a strongly regular graph (SRG) solution of the L-DUFM problem (1) is obtained by setting the matrices $(H_1, W_1, \ldots, W_L)$ as follows. For all $l$, the feature matrices $H_l, \tilde{H}_l$ are DNC1 collapsed, i.e., $H_l = M_l \otimes \mathbf{1}_n^T, \tilde{H}_l = \tilde{M}_l \otimes \mathbf{1}_n^T$. For $2 \leq l \leq L - 1$, $M_l = \tilde{M}_l$ and $M_l = A_l T_r$, where each row of $A_l$ is a multiple of the standard basis vector of dimension $r$ and the sum of squared multiples corresponding to one basis vector equals the same parameter $\alpha_l$. In other words, each row of $M_l$ is a multiple of a row of $T_r$ and the sum of squared norms of the rows of $M_l$ corresponding to a row of $T_r$ is the same for each row of $T_r$. For $l = 1$, $W_1, M_1$ are any pair of matrices that minimize the objective conditionally on $M_2$ being defined as above. For $l \geq 2$, $W_l$ minimizes the objective conditional to the input and output to that layer. Let $A_L^{(1)}$ be a $K \times r$ matrix where each row has exactly two $(-1)$ entries and exactly $r - 2$ $(+1)$ entries. Then, normalize the matrix $A_L^{(1)} T_r$ so that each row is unit norm and multiply from the left with $A_L^{(2)}$ of dimension $d_L \times K$, where each of the rows of $A_l^{(2)}$ is a multiple of a standard basis vector of dimension $K$ and the total sum of squared multiples corresponding to each basis vector is $\alpha_L$. Then, $\tilde{M}_L$ is obtained via this procedure, and $M_L = \sigma(\tilde{M}_L)$. Finally, the parameters $\{\alpha_l\}_{l=2}^L$ satisfy*

$$
\begin{aligned}
\alpha_l &= \frac{\left(\sqrt{n\lambda_{H_1}\lambda_{W_1}}\left(\sqrt{2} + \sqrt{(r-1)(r-2)}\right)\right)^{l-2}}{r^{l-2}\prod_{i=2}^{l-1}\lambda_{W_i}} q^l, \qquad 2 \leq l \leq L - 1, \\
\alpha_L &= \frac{\left(\sqrt{n\lambda_{H_1}\lambda_{W_1}}\left(\sqrt{2} + \sqrt{(r-1)(r-2)}\right)\right)^{L-2} 4((r-2)(r-3)+2)}{r^{L-1}(r-1)^2 \prod_{i=2}^{L-1}\lambda_{W_i}} q^L,
\end{aligned}
\tag{6}
$$

*and the parameter $q \geq 0$ is chosen to minimize the objective function in (1).*

**Definition 10.** *A deep neural collapse (DNC) solution for any number of classes $K$ of the L-DUFM problem (1) is obtained by setting the matrices $(H_1, W_1, \ldots, W_L)$ as follows. For all $l$, the feature matrices $H_l, \tilde{H}_l$ are DNC1 collapsed, i.e., $H_l = M_l \otimes \mathbf{1}_n^T, \tilde{H}_l = \tilde{M}_l \otimes \mathbf{1}_n^T$. For $2 \leq l \leq L$, $M_l = \tilde{M}_l$ and $M_l^T M_l = \alpha_l I_K$. For $l = 1$, $W_1, M_1$ are any pair of matrices that minimize the objective conditionally on $M_2$ being defined as above. For $l \geq 2$, $W_l$ minimizes the objective conditional to the input and output to that layer. Finally, the parameters $\{\alpha_l\}_{l=2}^L$ satisfy*

$$
\begin{aligned}
\alpha_{l-1} &= \frac{\lambda_{W_{L-1}}^{l-1}}{\lambda_{H_1} n \prod_{i=1}^{l-1}\lambda_{W_i}} q^{l-1}, \qquad 2 \leq l \leq L - 1, \\
\alpha_L &= \frac{\lambda_{W_{L-1}}^{L-1}}{\lambda_{H_1} n \prod_{i=1}^{L-2}\lambda_{W_i}} q^L,
\end{aligned}
$$

*and the parameter $q \geq 0$ is chosen to minimize the objective function in (1).*

Next, we define the SRG solution when $K \neq \binom{r}{2}$ for any $r$, and provide two constructions, each useful for different parts of the proof of Theorem 5.

**Definition 11.** *A strongly regular graph (SRG) solution for $K \geq 6$ of the L-DUFM problem (1), is obtained in one of the two following ways.*

1. *First, we take the largest $r$ s.t. $K \geq \binom{r}{2}$ and construct the SRG solution $(\bar{H}_1, \bar{W}_1, \ldots, \bar{W}_L)$ as in Definition 9 setting the number of classes to $\binom{r}{2}$. Next, we construct a DNC solution $(\tilde{H}_1, \tilde{W}_1, \ldots, \tilde{W}_L)$ as in Definition 10 setting the number of classes to $K - \binom{r}{2}$. Then, to construct $H_1$ of the SRG solution, we create it as a diagonal block matrix with number of columns equal to $K$ and number of rows equal to $\bar{d}_1 + \tilde{d}_1$, where these are the numbers of rows of the respective $H$ matrices; the first block is $\bar{H}_1$, the second block is $\tilde{H}_1$[2], and*

---

[2]The order is not important, both the rows and the columns can afterward be permuted if we accordingly permute also the weight matrices.

*the off-diagonal blocks are zero matrices. Similarly, we extend the weight matrices such that, for any l, $W_l$ is a block diagonal matrix where the number of rows is $\tilde{d}_{l+1} + \bar{d}_{l+1}$ and the number of columns is $\tilde{d}_l + \bar{d}_l$; the first block is $\bar{W}_l$, the second block is $\tilde{W}_l$, and the off-diagonal blocks are zero matrices.*

2. *First, we take the smallest $r$ s.t. $K \leq \binom{r}{2}$ and construct the SRG solution as in Definition 9 setting the number of classes to $\binom{r}{2}$. Then, we just remove the $\binom{r}{2} - K$ columns of $H_1$ that achieve the highest individual fit losses (in case of a tie choose arbitrarily), and define the SRG solution as the original solution without these columns.*

We recall our main result and give the proof.

**Theorem 5.** *If $K \geq 6, L \geq 4$ or $K \geq 10, L = 3$ and $d_l \geq K$ for all $l$, then $\mathcal{L}_{SRG} < \mathcal{L}_{DNC}$. Moreover, consider any sequence of L-DUFM problems for which $K \to \infty$ so that $0.499 > \mathcal{L}_{DNC}$ for each problem. In that case,*

$$\frac{\mathcal{L}_{SRG}}{\mathcal{L}_{DNC}} = \mathcal{O}(K^{\frac{3-L}{2(L+1)}}). \tag{3}$$

*Proof.* We start by considering the case $K = \binom{r}{2}$ for some $r$. Without loss of generality we can assume $n = 1$, because all comparisons are between solutions that are by definition DNC1 collapsed, and the ratio $\mathcal{L}_{SRG}/\mathcal{L}_{DNC}$ in the theorem statement does not depend on $n$.

We first compute the loss of the SRG solution as in Definition 9 up to only one degree of freedom. Let us go term-by-term. The simplest to evaluate is $\frac{\lambda_{W_l}}{2} \|W_l\|_F^2$ for $2 \leq l \leq L - 2$. Using Lemma 13 (which relies on Lemma 12), we get

$$\frac{\lambda_{W_l}}{2} \|W_l\|_F^2 = \frac{r\lambda_{W_l}}{2} \frac{\alpha_{l+1}}{\alpha_l}.$$

Similarly, for layer $L - 1$, we use Lemma 14 (again relying on Lemma 12) to compute:

$$\frac{\lambda_{W_{L-1}}}{2} \|W_{L-1}\|_F^2 = \frac{r^2(r-1)^2 \lambda_{W_{L-1}}}{8((r-2)(r-3)+2)} \frac{\alpha_L}{\alpha_{L-1}}.$$

Combining Lemma 15 with Lemma 18 we get:

$$\frac{\lambda_{W_1}}{2} \|W_1\|_F^2 + \frac{\lambda_{H_1}}{2} \|H_1\|_F^2 = \sqrt{\lambda_{W_1}\lambda_{H-1}} \left(\sqrt{2} + \sqrt{(r-1)(r-2)}\right) \alpha_2^{\frac{1}{2}}.$$

Finally, combining Lemma 17 with Lemma 16 we get:

$$\frac{1}{2K} \|W_L M_L - I_K\|_F^2 + \frac{\lambda_{W_L}}{2} \|W_L\|_F^2 = \frac{\lambda_{W_L}}{2} \frac{1}{\frac{(r-2)(5r-19)}{(r-2)(r-3)+2}\alpha_L + \frac{r(r-1)}{2}\lambda_{W_L}}$$

$$+ \frac{\lambda_{W_L}}{2} \frac{r-1}{\frac{2(r-3)^2}{(r-2)(r-3)+2}\alpha_L + \frac{r(r-1)}{2}\lambda_{W_L}}$$

$$+ \frac{\lambda_{W_L}}{2} \frac{\frac{r(r-3)}{2}}{\frac{2}{(r-2)(r-3)+2}\alpha_L + \frac{r(r-1)}{2}\lambda_{W_L}}.$$

The total loss of the SRG solution is just the sum of all these terms, which is expressed in terms of $\alpha_2, \alpha_3, \ldots, \alpha_L$. We now verify that the choice in (6) minimizes the loss, having set $q \equiv \alpha_2^{1/2}$. To do so, we compute the partial derivatives of $\mathcal{L}$ w.r.t. the $\alpha_l$'s and set them to 0:

$$0 = \frac{\partial \mathcal{L}}{\partial \alpha_2} = -r\lambda_{W_2}\frac{\alpha_3}{\alpha_2^2} + \sqrt{\lambda_{W_1}\lambda_{H-1}}\left(\sqrt{2} + \sqrt{(r-1)(r-2)}\right)\alpha_2^{-\frac{1}{2}} \iff$$

$$\frac{\alpha_3}{\alpha_2} = \frac{\sqrt{\lambda_{W_1}\lambda_{H-1}}\left(\sqrt{2} + \sqrt{(r-1)(r-2)}\right)}{r\lambda_{W_2}}q.$$

For $3 \leq l \leq L - 2$, we have

$$0 = \frac{\partial \mathcal{L}}{\partial \alpha_l} = -\frac{r\lambda_{W_l}}{2}\frac{\alpha_{l+1}}{\alpha_l^2} + \frac{r\lambda_{W_{l-1}}}{2}\frac{1}{\alpha_{l-1}} \iff$$

$$\frac{\alpha_{l+1}}{\alpha_l} = \frac{\lambda_{W_{l-1}}}{\lambda_{W_l}}\frac{\alpha_l}{\alpha_{l-1}}.$$

And finally, for layer $L - 1$, we have

$$0 = \frac{\partial \mathcal{L}}{\partial \alpha_{L-1}} = -\frac{\lambda_{W_{L-1}}}{2} \frac{r^2(r-1)^2}{4((r-2)(r-3)+2)} \frac{\alpha_L}{\alpha_{L-1}^2} + \frac{r\lambda_{W_{L-2}}}{2} \frac{1}{\alpha_{L-2}} \iff$$

$$\frac{\alpha_L}{\alpha_{L-1}} = \frac{\lambda_{W_{L-2}}}{\lambda_{W_{L-1}}} \frac{4((r-2)(r-3)+2)}{r(r-1)^2} \frac{\alpha_{L-1}}{\alpha_{L-2}}.$$

Denoting by $p = \sqrt{\lambda_{W_1}\lambda_{H_1}} \left( \sqrt{2} + \sqrt{(r-1)(r-2)} \right)$, we can express these fractions as

$$\frac{\alpha_{l+1}}{\alpha_l} = \frac{p}{r\lambda_{W_l}} q, \qquad 2 \le l \le L-2,$$

$$\frac{\alpha_L}{\alpha_{L-1}} = \frac{4((r-2)(r-3)+2)p}{r^2(r-1)^2\lambda_{W_{L-1}}} q,$$

which gives the expressions in (6). Finally, plugging this back into the loss function we get a univariate $q$-dependent function of the following form:

$$\mathcal{L}_{SRG}(q) = \frac{\lambda_{W_L}}{2} \frac{1}{\frac{4(r-2)(5r-19)\left(\sqrt{\lambda_{W_1}\lambda_{H_1}}\left(\sqrt{2}+\sqrt{(r-1)(r-2)}\right)\right)^{L-2}}{r^{L-1}(r-1)^2\prod_{i=2}^{L-1}\lambda_{W_i}} q^L + \frac{r(r-1)}{2}\lambda_{W_L}}$$

$$+ \frac{\lambda_{W_L}}{2} \frac{r-1}{\frac{8(r-3)^2\left(\sqrt{\lambda_{W_1}\lambda_{H_1}}\left(\sqrt{2}+\sqrt{(r-1)(r-2)}\right)\right)^{L-2}}{r^{L-1}(r-1)^2\prod_{i=2}^{L-1}\lambda_{W_i}} q^L + \frac{r(r-1)}{2}\lambda_{W_L}}$$

$$+ \frac{\lambda_{W_L}}{2} \frac{\frac{r(r-3)}{2}}{\frac{8\left(\sqrt{\lambda_{W_1}\lambda_{H_1}}\left(\sqrt{2}+\sqrt{(r-1)(r-2)}\right)\right)^{L-2}}{r^{L-1}(r-1)^2\prod_{i=2}^{L-1}\lambda_{W_i}} q^L + \frac{r(r-1)}{2}\lambda_{W_L}}$$

$$+ \frac{L}{2}\sqrt{\lambda_{W_1}\lambda_{H_1}}\left(\sqrt{2}+\sqrt{(r-1)(r-2)}\right) q.$$

The loss of the DNC solution can be computed by a simple extension of the expression (17) from [48]:

$$\mathcal{L}_{DNC}(q) = \frac{\lambda_{W_L}}{2} \frac{\frac{r(r-1)}{2}}{\frac{\lambda_{W_{L-1}}^{L-1}}{\lambda_{H_1}\prod_{i=1}^{L-2}\lambda_{W_i}} q^L + \frac{r(r-1)}{2}\lambda_{W_L}} + \frac{L}{2}\frac{r(r-1)}{2}\lambda_{W_{L-1}} q.$$

At this point, we split our analysis for $L = 3$ and for $L > 3$. We start with $L > 3$, which is simpler.

**Analysis for $L > 3$.** As $K \ge 6$, the following upper bound holds:

$$\mathcal{L}_{SRG}(q) \le \bar{\mathcal{L}}_{SRG}(q) := \frac{\lambda_{W_L}}{2} \frac{\frac{r(r-1)}{2}}{\frac{8\left(\sqrt{\lambda_{W_1}\lambda_{H_1}}\left(\sqrt{2}+\sqrt{(r-1)(r-2)}\right)\right)^{L-2}}{r^{L-1}(r-1)^2\prod_{i=2}^{L-1}\lambda_{W_i}} q^L + \frac{r(r-1)}{2}\lambda_{W_L}}$$

$$+ \frac{L}{2}\sqrt{\lambda_{W_1}\lambda_{H_1}}\left(\sqrt{2}+\sqrt{(r-1)(r-2)}\right) q.$$

Now we reparametrize $\mathcal{L}_{DNC}(q)$ and the upper bound on $\mathcal{L}_{SRG}(q)$, so that they look as similar as possible. By replacing $\lambda_{W_{L-1}} q$ with $q$, we get

$$\min_{q\ge 0} \mathcal{L}_{DNC}(q) = \min_{q\ge 0} \frac{\lambda_{W_L}}{2} \frac{\frac{r(r-1)}{2}}{\frac{1}{\lambda_{H_1}\prod_{i=1}^{L-1}\lambda_{W_i}} q^L + \frac{r(r-1)}{2}\lambda_{W_L}} + \frac{L}{2}r\frac{(r-1)}{2} q. \tag{7}$$

Similarly, by replacing $\frac{\sqrt{\lambda_{W_1}\lambda_{H_1}}\left(\sqrt{2}+\sqrt{(r-1)(r-2)}\right)}{r} q$ with $q$, we get:

$$\min_{q\ge 0} \bar{\mathcal{L}}_{SRG}(q) = \min_{q\ge 0} \frac{\lambda_{W_L}}{2} \frac{\frac{r(r-1)}{2}}{\frac{8r}{\left(\sqrt{2}+\sqrt{(r-1)(r-2)}\right)^2(r-1)^2\lambda_{H_1}\prod_{i=1}^{L-1}\lambda_{W_i}} q^L + \frac{r(r-1)}{2}\lambda_{W_L}}$$

$$+ \frac{L}{2}rq.$$

Next, by replacing $\left( \frac{8r}{\left(\sqrt{2}+\sqrt{(r-1)(r-2)}\right)^2 (r-1)^2} \right)^{\frac{1}{L}} q$ with $q$, we get:

$$\min_{q\geq 0} \bar{\mathcal{L}}_{SRG}(q) = \min_{q\geq 0} \frac{\lambda_{W_L}}{2} \frac{\frac{r(r-1)}{2}}{\frac{1}{\lambda_{H_1} \prod_{i=1}^{L-1} \lambda_{W_i}} q^L + \frac{r(r-1)}{2}\lambda_{W_L}} \tag{8}$$

$$+ \frac{L}{2} r \left( \frac{\left(\sqrt{2} + \sqrt{(r-1)(r-2)}\right)^2 (r-1)^2}{8r} \right)^{\frac{1}{L}} q.$$

After the reparameterization, $\mathcal{L}_{DNC}(q)$ and $\bar{\mathcal{L}}_{SRG}(q)$ have almost the same form except for the multiplier $\frac{r-1}{2}$ in the DNC case and

$$\left( \frac{\left(\sqrt{2} + \sqrt{(r-1)(r-2)}\right)^2 (r-1)^2}{8r} \right)^{\frac{1}{L}}$$

in the SRG case. Therefore, the inequality between $\mathcal{L}_{DNC}$ and $\bar{\mathcal{L}}_{SRG}$ is fully determined by the inequality between these two terms. We can write:

$$\frac{r-1}{2} > \left( \frac{\left(\sqrt{2} + \sqrt{(r-1)(r-2)}\right)^2 (r-1)^2}{8r} \right)^{\frac{1}{L}} \iff$$

$$r(r-1)^{L-2} > 2^{L-3} \left(\sqrt{2} + \sqrt{(r-1)(r-2)}\right)^2$$

We first solve it for $L = 4$ and any $r \geq 4$ (which is guaranteed by $K \geq 6$). We get the inequality $r(r-1)^2 > 2(r-1)(r-2) + 4 + 4\sqrt{2(r-1)(r-2)}$. This inequality is equivalent to $(r-1)(r^2 - 3r + 4) > 4 + 4\sqrt{2(r-1)(r-2)}$, which holds for all $r \geq 4$.

Compared to the case $L = 4$, for general $L$ the LHS gets multiplied by $(r-1)^{L-4}$ and the RHS gets multiplied by $2^{L-4}$, which is smaller for $r \geq 4$. Hence, the inequality holds as well.

**Analysis for $L = 3$.** Here, we need a tighter upper bound than $\bar{\mathcal{L}}_{SRG}(q)$. Thus, we write

$$\mathcal{L}_{SRG}(q) \leq \frac{\lambda_{W_3}}{2} \frac{r}{\frac{8(r-3)^2\sqrt{\lambda_{W_1}\lambda_{H_1}}\left(\sqrt{2}+\sqrt{(r-1)(r-2)}\right)}{r^2(r-1)^2\lambda_{W_2}} q^3 + \frac{r(r-1)}{2}\lambda_{W_3}}$$

$$+ \frac{\lambda_{W_3}}{2} \frac{\frac{r(r-3)}{2}}{\frac{8\sqrt{\lambda_{W_1}\lambda_{H_1}}\left(\sqrt{2}+\sqrt{(r-1)(r-2)}\right)}{r^{L-1}(r-1)^2\lambda_{W_2}} q^3 + \frac{r(r-1)}{2}\lambda_{W_3}}$$

$$+ \frac{3}{2}\sqrt{\lambda_{W_1}\lambda_{H_1}} \left(\sqrt{2} + \sqrt{(r-1)(r-2)}\right) q.$$

We equivalently re-write this by extending both of the ratios by $\frac{r-2}{r-2}\frac{r-1}{r-1}$ and then moving $\frac{r-2}{r-1}$ to denominator. Thus,

$$\mathcal{L}_{SRG}(q) \leq \tilde{\mathcal{L}}_{SRG}(q) := \frac{\lambda_{W_3}}{2} \frac{\frac{r(r-1)}{r-2}}{\frac{8(r-3)^2\sqrt{\lambda_{W_1}\lambda_{H_1}}\left(\sqrt{2}+\sqrt{(r-1)(r-2)}\right)}{r^2(r-1)(r-2)\lambda_{W_2}} q^3 + \frac{r(r-1)}{2}\lambda_{W_3}}$$

$$+ \frac{\lambda_{W_3}}{2} \frac{\frac{r(r-3)(r-1)}{2(r-2)}}{\frac{8\sqrt{\lambda_{W_1}\lambda_{H_1}}\left(\sqrt{2}+\sqrt{(r-1)(r-2)}\right)}{r^2(r-1)(r-2)\lambda_{W_2}} q^3 + \frac{r(r-1)}{2}\lambda_{W_3}}$$

$$+ \frac{3}{2}\sqrt{\lambda_{W_1}\lambda_{H_1}} \left(\sqrt{2} + \sqrt{(r-1)(r-2)}\right) q.$$

Now, we perform the same reparametrizations as for $L > 4$, with the only exception of treating $(r-1)(r-2)$ in the denominator of the denominator of the current ratios as $(r-1)^2$ in the previous case. Then, we have

$$
\min_{q \geq 0} \tilde{\mathcal{L}}_{SRG}(q) = \min_{q \geq 0} \frac{\lambda_{W_3}}{2} \frac{\frac{r(r-1)}{r-2}}{\frac{(r-3)^2}{\lambda_{H_1} \lambda_{W_1} \lambda_{W_2}} q^3 + \frac{r(r-1)}{2} \lambda_{W_3}}
$$
$$
+ \frac{\lambda_{W_3}}{2} \frac{\frac{r(r-3)(r-1)}{2(r-2)}}{\frac{1}{\lambda_{H_1} \lambda_{W_1} \lambda_{W_2}} q^3 + \frac{r(r-1)}{2} \lambda_{W_3}}
$$
$$
+ \frac{3}{2} r \left( \frac{\left( \sqrt{2} + \sqrt{(r-1)(r-2)} \right)^2 (r-1)(r-2)}{8r} \right)^{\frac{1}{3}} q.
$$

Assume that the following inequality holds:

$$
\frac{1}{\frac{(r-3)^2}{\lambda_{H_1} \lambda_{W_1} \lambda_{W_2}} q^3 + \frac{r(r-1)}{2} \lambda_{W_3}} \leq \frac{1}{2} \frac{1}{\frac{1}{\lambda_{H_1} \lambda_{W_1} \lambda_{W_2}} q^3 + \frac{r(r-1)}{2} \lambda_{W_3}}. \tag{9}
$$

Then,

$$
\min_{q \geq 0} \tilde{\mathcal{L}}_{SRG}(q) \leq \min_{q \geq 0} \bar{\mathcal{L}}_{SRG}(q) := \min_{q \geq 0} \frac{\lambda_{W_3}}{2} \frac{\frac{r(r-1)}{2}}{\frac{1}{\lambda_{H_1} \lambda_{W_1} \lambda_{W_2}} q^3 + \frac{r(r-1)}{2} \lambda_{W_3}}
$$
$$
+ \frac{3}{2} r \left( \frac{\left( \sqrt{2} + \sqrt{(r-1)(r-2)} \right)^2 (r-1)(r-2)}{8r} \right)^{\frac{1}{3}} q.
$$

The only difference between the expression considered here and the one considered in the $L > 3$ case is that here we have $(r-1)(r-2)$ instead of $(r-1)^2$ within the expression. By comparing against $\mathcal{L}_{DNC}$ again, we get that $\bar{\mathcal{L}}_{SRG} < \mathcal{L}_{DNC}$ if and only if $r(r-1)^2 > \left( \sqrt{2} + \sqrt{(r-1)(r-2)} \right)^2 (r-2)$. This is equivalent to $(r-1)(3r-4) > 2(r-2) + 2(r-2)\sqrt{2(r-1)(r-2)}$, which holds for all $r \geq 4$.

It remains to show that (9) holds, and it suffices to do so for the minimizer $q^*$ of $\mathcal{L}_{DNC}$, as $\min_{q \geq 0} \tilde{\mathcal{L}}_{SRG}(q) \leq \tilde{\mathcal{L}}_{SRG}(q^*) \leq \bar{\mathcal{L}}_{SRG}(q^*) < \mathcal{L}_{DNC}(q^*) = \min_{q \geq 0} \mathcal{L}_{DNC}(q)$. Note that this is equivalent to

$$
\frac{r(r-1)}{2} \lambda_{W_3} < \frac{(r-3)^2 - 2}{\lambda_{W_1} \lambda_{W_2} \lambda_{H_1}} (q^*)^3.
$$

Note that the minimum of the function in (7) (having the same reparametrization as $\bar{\mathcal{L}}_{SRG}$) – if it is not at $q^* = 0$, in which case the statement of the theorem is trivial – must come after the unique inflection point of the function. A direct computation yields that this inflection point satisfies

$$
\frac{r(r-1)}{2} \lambda_{W_3} = \frac{2}{\lambda_{W_1} \lambda_{W_2} \lambda_{H_1}} q^3.
$$

Therefore, the minimum of (7) is attained at $q^*$ for which

$$
\frac{r(r-1)}{2} \lambda_{W_3} < \frac{2}{\lambda_{W_2} \lambda_{W_1} \lambda_{H_1}} (q^*)^3.
$$

For $r \geq 5$, this implies that such a $q^*$ satisfies (9), which concludes the argument for $K = \binom{r}{2}$.

For a general $K$, the extension is rather simple. Note that the *first type of* SRG solution in Definition 11 is constructed in a way so that the losses attained by the SRG and DNC parts sum up. Therefore, we can split the analysis for the SRG and DNC parts. The DNC part obviously attains equal loss to the DNC solution. For the SRG part, the analysis done above applies, and the argument is complete.

It remains to show the statement on the asymptotic relationship between $\mathcal{L}_{SRG}$ and $\mathcal{L}_{DNC}$ for $K \to \infty$ when $L \geq 4$. Formally, we should consider sequences of the problems and label everything with an extra index corresponding to the order within the sequence. However, with an abuse of notation, we drop this indexing and switch to the $\mathcal{O}, \Theta$ notations whenever convenient.

As before, we start by considering $K$ of the form $\binom{r}{2}$ for some $r$. Let $\Lambda = \lambda_{H_1} \prod_{i=1}^{L} \lambda_{W_i}$ and

$$\Psi(K) = \left( \frac{2^{L-3} \left( \sqrt{2} + \sqrt{(r-1)(r-2)} \right)^2}{r(r-1)^{L-2}} \right)^{\frac{1}{L}}, \text{ where } r \text{ corresponds to the value s.t. } \binom{r}{2} = K. \text{ We note}$$

that $\Psi(K) = \Theta(K^{\frac{3-L}{2L}})$. Since we are interested in the ratio $\frac{\mathcal{L}_{SRG}}{\mathcal{L}_{DNC}}$, we do a few changes and reparametrizations to the expressions in (7) and (8): we multiply both by 2, divide all terms in the left summands by $\lambda_{W_L}$, rewrite $\frac{r(r-1)}{2}$ as $K$, plug in the defined quantities, divide all the terms in the left summands by $K$ and finally replace $\Lambda^{-\frac{1}{L}} K^{-\frac{1}{L}} q$ with $q$ to obtain the following expression

$$\frac{1}{q^L + 1} + LK^{\frac{L+1}{L}} \Lambda^{\frac{1}{L}} q \tag{10}$$

for the DNC loss, and the following expression

$$\frac{1}{q^L + 1} + LK^{\frac{L+1}{L}} \Lambda^{\frac{1}{L}} \Psi(K) q \tag{11}$$

for the SRG loss. Using a similar trick as in the previous analysis for $L = 3$, we have that the minimum of the function in (10) is achieved when $q > 1$. Hence, we can lower bound (10) by

$$\frac{1}{3} q^{-L} + LK^{\frac{L+1}{L}} \Lambda^{\frac{1}{L}} q.$$

For this convex expression, we can find the optimal solution by setting to zero the derivative, which gives that the optimal solution is $(1 + L) 3^{-\frac{1}{L+1}} K \Lambda^{\frac{1}{L+1}}$. Similarly, we can upper bound (11) by

$$q^{-L} + LK^{\frac{L+1}{L}} \Lambda^{\frac{1}{L}} \Psi(K) q$$

and after finding the optimal solution we get that it equals $(1 + L) K \Lambda^{\frac{1}{L+1}} \Psi(K)^{\frac{L}{L+1}}$. This allows us to conclude that

$$\frac{\mathcal{L}_{SRG}}{\mathcal{L}_{DNC}} = \mathcal{O}(K^{\frac{3-L}{2(L+1)}}).$$

To get the same formula when the number of classes is not of the form $\binom{r}{2}$, we only need simple adjustments. For this part, we will employ the upper-index notation to denote the number of classes $K$ to which the solution corresponds. First, note that the optimal value of (10) is continuous in the coefficient in front of the linear term $q$. Therefore, if $\mathcal{L}_{DNC}^K < 0.499$, then, choosing the smallest $r$ for which $K \leq \binom{r}{2} := \bar{K}$, we see that $\mathcal{L}_{DNC}^{\bar{K}} < 0.5$ for the same set of regularization parameters, as $\frac{\bar{K}}{K} \xrightarrow{K \to \infty} 1$. Since the argument above does not need $\mathcal{L}_{DNC} < 0.499$ but only $\mathcal{L}_{DNC} < 0.5$, we can now use that $\mathcal{L}_{SRG}^{\bar{K}}$ with the same regularization parameters is still $\mathcal{O}(K^{\frac{3-L}{2(L+1)}})$. Finally, choosing the second construction in Definition 11, we construct the SRG solution for $K$ classes from the SRG solution for $\bar{K}$ classes with the same regularization parameters (thus also the same regularization as for the DNC solution with $K$ classes). To conclude, it just suffices to see that $\mathcal{L}_{SRG}^{\bar{K}} \geq \mathcal{L}_{SRG}^K$ because we removed columns from $H_1$, decreasing its norm and the fit loss is at most as big because the columns with the worst fit loss were removed and the fit loss is an average over the columns. This concludes the proof also for general $K$. $\qquad \square$

We conclude the section by stating and proving a few auxiliary lemmas that were used in the proof of Theorem 5.

**Lemma 12.** *Consider the following optimization problem:*

$$\min_w \|w\|^2 \tag{12}$$

$$\text{s.t. } z^T = w^T A_l T_r. \tag{13}$$

*Then, the value of the optimal solution is*

$$\frac{(r-1)^2}{\alpha_l (r-2)^2} z^T T_r^T \left( I_r - \frac{3r-4}{4(r-1)^2} \mathbf{1}_r \mathbf{1}_r^T \right) T_r z.$$

*Proof.* Multiplying the constraint with $T_r^T(T_r T_r^T)^{-1}$ from the right we get $z^T T_r^T(T_r T_r^T)^{-1} = w^T A_l$. Now, we can use that the minimum $l_2$ norm solution of such a system can be computed by multiplying with the right pseudoinverse of $A_l$. Thus, we get $w = A_l(A_l^T A_l)^{-1}(T_r T_r^T)^{-1} T_r z = 1/\alpha_l A_l(T_r T_r^T)^{-1} T_r z$. Then the squared norm of this is simply:

$$w^T w = \frac{1}{\alpha_l^2} z^T T_r^T (T_r T_r^T)^{-1} A_l^T A_l (T_r T_r^T)^{-1} T_r z = \frac{1}{\alpha_l} z^T T_r^T (T_r T_r^T)^{-2} T_r z.$$

Now, we know that

$$T_r T_r^T = \frac{r-2}{r-1} I_r + \frac{1}{r-1} \mathbf{1} \mathbf{1}^T = \frac{r-2}{r-1} \left( I + \frac{1}{r-2} \mathbf{1} \mathbf{1}^T \right).$$

This can be seen by looking at the structure of $\mathcal{K}_n$ where two vertices have exactly one edge between them. Now we can compute the the inverse of this matrix using the Sherman-Morrison formula

$$\left( I + \frac{1}{r-2} \mathbf{1} \mathbf{1}^T \right)^{-1} = I - \frac{1}{2(r-1)} \mathbf{1} \mathbf{1}^T$$

and the square is:

$$\left( I + \frac{1}{r-2} \mathbf{1} \mathbf{1}^T \right)^{-2} = I - \frac{3r-4}{4(r-1)^2} \mathbf{1} \mathbf{1}^T.$$

Putting this all together, the proof is complete. $\square$

**Lemma 13.** *Let $2 \le l \le L-2$. Consider $\tilde{M}_{l+1}, M_l$ as in Definition 9 of the SRG solution. Then, the following optimization problem:*

$$\min_W \|W\|_F^2 \tag{14}$$

$$s.t. \ \tilde{M}_{l+1} = W M_l \tag{15}$$

*achieves optimal value of $\frac{r\alpha_{l+1}}{\alpha_l}$.*

*Proof.* This is a direct consequence of Lemma 12. Denote $\gamma z_0$ a row from $\tilde{M}_{l+1}$ which corresponds to the first row of $T_r$ and such that $\|z_0\| = 1$. Then, $T_r \gamma z_0 = \gamma(1, 1/(r-1), \ldots, 1/(r-1))^T$. Directly evaluating the expression in Lemma 12 will yield $\gamma^2 \alpha_l^{-1}$ for a single row and thus for all rows we get the value from the lemma statement. $\square$

**Lemma 14.** *Consider $\tilde{M}_L, M_{L-1}$ as in Definition 9 of the SRG solution. Then, the following optimization problem:*

$$\min_W \|W\|_F^2 \tag{16}$$

$$s.t. \ \tilde{M}_L = W M_{L-1} \tag{17}$$

*achieves optimal value of*

$$\frac{\alpha_L}{\alpha_{L-1}} \frac{r^2(r-1)^2}{4((r-2)(r-3)+2)}.$$

*Proof.* We first need to characterize the rows of $\tilde{M}_L$. Note that $\tilde{M}_L$ comes from the multiplication of $T_r$ and $A_L^{(1)}$, see Definition 9. This operation can be seen as weighting vertices of $\mathcal{K}_r$ (rows of $T_r$) and then looking at what the sum of the weights of adjacent vertices of each edge (column of $T_r$) is. We can easily see that the resulting vector has exactly one "$-1$" entry (for the edge corresponding to the two vertices given negative weight) and exactly $\binom{r-2}{2}$ entries with "$+1$". Therefore, it must be scaled with the inverse of $s := \sqrt{\frac{(r-2)(r-3)}{2} + 1}$ to be unit norm. Now, let $z_1$ be one of these vectors with the negative edge between first two vertices. Then,

$$T_r z_1 = \left( -\frac{1}{s\sqrt{r-1}}, -\frac{1}{s\sqrt{r-1}}, \frac{r-3}{s\sqrt{r-1}}, \frac{r-3}{s\sqrt{r-1}}, \ldots \right)^T,$$

which can be easily derived if we imagine doing edge-wise dot-product between two vertex weightings, one with two "−1s" for $z_1$ and the other type with one "+1" representing the rows of $T_r$. For the other vectors in $\tilde{H}_L$, the resulting vectors would be similar, except they would have the negative entries for different pairs of vertices of $\mathcal{K}_r$. Note that

$$z_1^T T_r^T T_r z = \frac{2}{s^2(r-1)} + \frac{(r-3)^2(r-2)}{s^2(r-1)},$$

$$z_1^T T_r^T \mathbf{1}_r \mathbf{1}_r^T T_r z = \frac{((r-3)(r-2)-2)^2}{s^2(r-1)}.$$

Therefore, if we are optimizing for $\gamma z_1$, then an application of Lemma 12 gives

$$w^T w = \gamma^2 \alpha_{L-1}^{-1} \frac{(r-1)^2}{(r-2)^2} \left( \frac{2}{s^2(r-1)} + \frac{(r-3)^2(r-2)}{s^2(r-1)} - \frac{3r-4}{4(r-1)^2} \frac{((r-3)(r-2)-2)^2}{s^2(r-1)} \right),$$

where $w$ denotes a row of $W$. Simplifying this expression, we get

$$w^T w = \gamma^2 \alpha_{L-1}^{-1} \frac{r(r-1)}{2((r-2)(r-3)+2)}.$$

To conclude, it suffices to sum up $\frac{r(r-1)}{2}$ such rows having total $l_2$ norm squared $\alpha_L$ (and, hence, $\gamma^2 = \alpha_L$), which gives

$$\|W\|_F^2 = \frac{\alpha_L}{\alpha_{L-1}} \frac{r^2(r-1)^2}{4((r-2)(r-3)+2)},$$

and concludes the proof. $\qquad\square$

**Lemma 15.** *For $2 \leq l \leq L-1$, consider $M_l, \tilde{M}_l$ as in Definition 9 of the SRG solution. Then, $M_l = \tilde{M}_l$ and the eigenvalues of $M_l^T M_l$ are:*

$$\mu_1 = 2\alpha_l \quad \text{with multiplicity } 1,$$

$$\mu_2 = \frac{r-2}{r-1}\alpha_l \quad \text{with multiplicity } r-1,$$

$$\mu_3 = 0 \quad \text{with multiplicity } \frac{r(r-3)}{2}.$$

*Proof.* From the definition, it readily follows that $M_l = \tilde{M}_l$, so let us compute $M_l^T M_l$. Looking at any row of $M_l$, we see that it has non-negative equal entries of value $\sqrt{\alpha_{ij}}/\sqrt{r-1}$, where $\sum_j \alpha_{ij} = \alpha_l$ on all the $r-1$ edges of $\mathcal{K}_r$ that contain the vertex corresponding to that row. Therefore, by definition of $\alpha_l$, the sum of squares of all entries corresponding to one row type within any column is $\alpha_l/(r-1)$. Each edge (and, thus, column) contains exactly two vertices, thus the diagonal elements of $M_l^T M_l$, which are the $l_2$ norms squared of the columns of $M_l$, are simply equal to $\frac{2\alpha_l}{r-1}$. There are two possible off-diagonal values. One is for the pairs of columns that correspond to edges that share a vertex and one is for those pairs that do not share a vertex. The pairs of columns whose edges do not share a vertex do not have any entries which would *both* be jointly positive, because either a vertex does not belong to one edge or to the other. Therefore, the value of off-diagonal entries corresponding to such pairs is simply 0. On the other hand, there is exactly one vertex that has non-zero values for *both* edges corresponding to columns whose edges do share a vertex – it is the shared vertex. Therefore the value of off-diagonal entries of this type is $\frac{\alpha_l}{r-1}$. Crucially, the structure of the off-diagonal entries is fully determined by the graph $\mathcal{T}_r$, because two edges in $\mathcal{K}_r$ share a vertex if and only if they are connected in the graph $\mathcal{T}_r$. Therefore, $M_l^T M_l$ can be written as a weighted sum of $I_K$ and the adjacency matrix $G_r$ of $\mathcal{T}_r$, where the weight of $I_K$ simply corresponds to the size of the diagonal term and the weight of $G_r$ to the positive off-diagonal term. In conclusion, we get

$$M_l^T M_l = \frac{2\alpha_l}{r-1} I_K + \frac{\alpha_l}{r-1} G_r.$$

As $\mathcal{T}_r$ is a strongly regular graph with parameters $(r(r-1)/2, 2(r-2), r-2, 4)$, $G_r$ has a single eigenvalue equal to $2(r-2)$, $r-1$ eigenvalues equal to $r-4$ and $r(r-3)/2$ eigenvalues equal to $-2$, which concludes the proof. $\qquad\square$

**Lemma 16.** *Consider $M_L$ as in Definition 9 of the SRG solution. Then, the eigenvalues of $M_L^T M_L$ are:*

$$\mu_1 = \frac{(r-2)(5r-19)\alpha_L}{(r-2)(r-3)+2} \quad \text{with multiplicity } 1,$$

$$\mu_2 = \frac{2(r-3)^2\alpha_L}{(r-2)(r-3)+2} \quad \text{with multiplicity } r-1,$$

$$\mu_3 = \frac{2\alpha_L}{(r-2)(r-3)+2} \quad \text{with multiplicity } \frac{r(r-3)}{2}.$$

*Proof.* Let us compute $M_L^T M_L$. Looking at any row of $M_L$, we see that it has non-negative equal entries of value $\sqrt{\alpha_{ij}}/s$, where $s = \sqrt{\frac{(r-2)(r-3)}{2}+1}$ and $\sum_j \alpha_{ij} = \alpha_L$ on all edges in a subgraph of $\mathcal{K}_r$ of size $r-2$. Therefore, by definition of $\alpha_L$, the sum of squares of all entries corresponding to one row type within any column is $\alpha_L/s^2$. However, not all row types have non-zero value on any particular column. Namely, a row type will only have non-zero value on a column, if the row-type corresponds to such a subgraph of $\mathcal{K}_r$, which is disjoint with the edge corresponding to the column. This is because all edges outside the complete subgraph of $r-2$ vertices corresponding to the row type are assigned 0 in $M_L$. Therefore, the number of row types that assign non-zero value in a particular column is equal to the number of $r-2$ vertex sets. This corresponds to the number of edges in $\mathcal{K}_r$, which is equal to $\binom{r-2}{2} = s^2 - 1$. Thus, the diagonal elements of $M_L^T M_L$, which are the $l_2$ norms squared of the columns of $M_L$, are simply equal to $\frac{(s^2-1)\alpha_L}{s^2}$. There are two possible off-diagonal values. One is for the pairs of columns that correspond to edges that share a vertex, and one is for those pairs that don't share a vertex. Let us compute the number of row types assigning positive value to *both* of these columns jointly. Using the same interpretation, the columns correspond to edges and only row types that correspond to $r-2$ vertex subsets disjoint with them assign positive value to the column of that edge. If we want this to be satisfied for both rows jointly, we need to take the intersection of those $r-2$ vertex subsets, which in this case will result in an $r-3$ vertex subset. Thus, exactly $\binom{r-3}{2}$ row types will jointly assign a positive value. Therefore, we have value $\frac{(r-3)(r-4)\alpha_L}{2s^2}$ on these off-diagonal entries. For the pairs of columns that correspond to edges with disjoint vertices, the same intersection will now yield a set of vertices of size only $r-4$. Therefore, the value of this off-diagonal entry is $\frac{(r-4)(r-5)\alpha_L}{2s^2}$. Crucially, the structure of the off-diagonal entries is fully determined by the graph $\mathcal{T}_r$, because two edges in $\mathcal{K}_r$ share a vertex if and only if they are connected in the graph $\mathcal{T}_r$. Therefore, $M_L^T M_L$ can be written as a weighted sum of $\mathbf{1}_K\mathbf{1}_K^T, I_K$ and the adjacency matrix $G_r$ of $\mathcal{T}_r$. The weights can be determined as follows: we first subtract a multiple of $\mathbf{1}_K\mathbf{1}_K^T$ to make the smaller off-diagonal entry of $M_L^T M_L$ zero, then we subtract what is left of the diagonal and we take the rest to be a multiple of $G_r$. In conclusion, we get

$$M_L^T M_L = \frac{(r-4)(r-5)\alpha_L}{2s^2}\mathbf{1}_K\mathbf{1}_K^T$$
$$+ \left(\frac{(s^2-1)\alpha_L}{s^2} - \frac{(r-4)(r-5)\alpha_L}{2s^2}\right) I_K$$
$$+ \left(\frac{(r-3)(r-4)\alpha_L}{2s^2} - \frac{(r-4)(r-5)\alpha_L}{2s^2}\right) G_r.$$

As $\mathcal{T}_r$ is a strongly regular graph with parameters $(r(r-1)/2, 2(r-2), r-2, 4)$, $G_r$ has a single eigenvalue equal to $2(r-2)$, $r-1$ eigenvalues equal to $r-4$ and $r(r-3)/2$ eigenvalues equal to $-2$. The summation with $I_K$ only shifts all the eigenvalues. The term $\mathbf{1}_K\mathbf{1}_K^T$ has only one non-zero eigenvalue, and the eigenvector is identical to that of the eigenvector corresponding to the dominant eigenvalue of $G_r$. This concludes the proof. □

**Lemma 17.** *Assuming DNC1, let $M_L$ be the mean matrix of the last layer. Let $M_L = U\Sigma V^T$ be the full SVD of $M_L$ and let $\sigma_i$, $i \in [K]$, be the singular values of $M_L$. Then, the following optimization problem:*

$$\min_{W_L} \frac{1}{2K} \|W_L M_L - I_K\|_F^2 + \frac{\lambda_{W_L}}{2} \|W_L\|_F^2$$

*attains the minimum of*

$$\frac{\lambda_{W_L}}{2} \sum_{i=1}^{K} \frac{1}{\sigma_i^2 + K\lambda_{W_L}}.$$

*Proof.* The proof consists in a direct computation. Let $W_L^*$ denote the minimizer. Computing the gradient and setting it to $0$ gives that

$$W_L^* = M_L^T(M_L M_L^T + \lambda_{W_L} K I_{d_L})^{-1} = V\Sigma^T(\Sigma\Sigma^T + \lambda_{W_L} K I_{d_L})^{-1}U^T,$$

which readily implies that

$$\|W_L^*\|_F^2 = \sum_{i=1}^{K} \frac{\sigma_i^2}{(\sigma_i^2 + \lambda_{W_L} K)^2},$$

$$\|W_L M_L - I_K\|_F^2 = \left\|V\Sigma^T(\Sigma\Sigma^T + \lambda_{W_L} K I_{d_L})^{-1}\Sigma V^T - VV^T\right\|_F^2$$

$$= \sum_{i=1}^{K} \left(\frac{\sigma_i^2}{\sigma_i^2 + \lambda_{W_L} K} - 1\right)^2 = \sum_{i=1}^{K} \frac{K^2\lambda_{W_L}^2}{(\sigma_i^2 + \lambda_{W_L} K)^2},$$

thus concluding the argument. $\qquad\square$

**Lemma 18.** *The optimization problem*

$$\min_{A,B;C=AB} \quad \frac{\lambda_A}{2}\|A\|_F^2 + \frac{\lambda_B}{2}\|B\|_F^2 \tag{18}$$

*attains the minimum of $\sqrt{\lambda_A\lambda_B}\,\|C\|_*$, and the minimizers are of the form $A^* = \gamma_A U\Sigma^{1/2}R^T, B^* = \gamma_B R\Sigma^{1/2}V^T$. Here, the constants $\gamma_A, \gamma_B$ only depend on $\lambda_A, \lambda_B$; $U\Sigma V^T$ is the SVD of $C$; and $R$ is an orthogonal matrix.*

*Proof.* See Lemma C.1 of [51]. $\qquad\square$

## A.2 No within-class variability is still optimal

**Theorem 6.** *The optimal solutions of the L-DUFM* (1) *exhibit DNC1 at layer $L$, i.e.,*

$$H_L^* = M_L^* \otimes \mathbf{1}_n^T$$

*holds for any optimal solution $(H_1^*, W_1^*, \ldots, W_L^*)$ of the L-DUFM problem.*

*Proof.* **Step 1: Reduction to $n = 1$.** In the first step, assume by contradiction that there exists an optimal solution of (1) with regularization parameters $(\lambda_{H_1}, \lambda_{W_1}, \ldots, \lambda_{W_L})$ denoted as $(H_1^*, W_1^*, \ldots, W_L^*)$ which does not exhibit deep neural collapse at layer $L$. This means that there exist indices $c, i, j$ s.t. $h_{ci}^L \neq h_{cj}^L$. Let us construct two solutions of the $n = 1$ L-DUFM. They will share the weight matrices which will equal $(W_1^*, \ldots, W_L^*)$ – the weight matrices of the original solution. To construct the features, for every class except the $c$-th, pick any sample and share it between both solutions. For the class $c$, take the samples $h_{ci}^L, h_{cj}^L$ and put one in one solution and the other one in the other solution. Denote $H_1^{(1)}, H_1^{(2)}$ the two $n = 1$ sample matrices. It is not hard to see that both $(H_1^{(1)}, W_1^*, \ldots, W_L^*)$ and $(H_1^{(2)}, W_1^*, \ldots, W_L^*)$ are optimal solutions of (1) with regularization parameters $(n\lambda_{H_1}, \lambda_{W_1}, \ldots, \lambda_{W_L})$. To prove it, assume by contradiction that, without loss of generality, $(H_1^{(2)}, W_1^*, \ldots, W_L^*)$ is not an optimal solution of the corresponding problem. Then, there exists an alternative $(H_1^{(0)}, \hat{W}_1^*, \ldots, \hat{W}_L^*)$ that achieves smaller loss for this problem. Let us duplicate all the samples of $\hat{H}_1^{(0)}$ for $n$ times, thus constructing $\hat{H}_1^* = H_1^{(0)} \otimes \mathbf{1}_n^T$. The solution $(\hat{H}_1^*, \hat{W}_1^*, \ldots, \hat{W}_L^*)$ has the same loss under the L-DUFM problem with regularization parameters $(\lambda_{H_1}, \lambda_{W_1}, \ldots, \lambda_{W_L})$ as the solution $(H_1^{(0)}, \hat{W}_1^*, \ldots, \hat{W}_L^*)$ for the L-DUFM with $n = 1$ and parameters $(n\lambda_{H_1}, \lambda_{W_1}, \ldots, \lambda_{W_L})$. This is easy to see from the separability of both $\|H_1\|_F^2$ and the fit part of the loss in (1) w.r.t. the columns of $H_1$. For the same reasons, the loss

functions for $(H_1^*, W_1^*, \ldots, W_L^*)$ in the original problem equals the loss function of the solutions $(H_1^{(1)}, W_1^*, \ldots, W_L^*)$ or $(H_1^{(2)}, W_1^*, \ldots, W_L^*)$ in the reduced problem. In fact, if this was not the case, there would need to be an inequality between the losses exhibited by two different columns of $H_1$ belonging to the same class, from which we could arrive at a contradiction by taking the better column and multiplying it to all columns within that class, thereby obtaining a better solution. This means that the loss of $(\hat{H}_1^*, \hat{W}_1^*, \ldots, \hat{W}_L^*)$ in the original problem is smaller than the loss of $(H_1^*, W_1^*, \ldots, W_L^*)$, which is a contradiction.

**Step 2: Excluding an aligned case.** By assumption we know that not only $H_1^{(1)}$ and $H_1^{(2)}$ differ in the $c$-th column (from now on we assume without loss of generality that it is the first column) but also $H_L^{(1)}$ and $H_L^{(2)}$ do. Denote for simplicity the first (differing) columns of $H_L^{(1)}$ and $H_L^{(2)}$ as $x, y$ respectively. We now show that it is not possible that $y = \alpha x$. First, $\alpha$ has to be non-negative since $x, y$ are entry-wise non-negative given that they come after the application of $\sigma$. Assume w.l.o.g. $\alpha > 1$ (otherwise, we can just exchange the roles of $x$ and $y$). Consider a reduced problem where we only optimize for the *size* of the first column of either $H_L^{(1)}$ or $H_L^{(2)}$, focusing on that part of the problem (1) which is relevant for this column, being:

$$\min_{t \geq 0} \frac{1}{2N} \left\| t W_L h_{11}^{L(i)} - e_1 \right\|_2^2 + \frac{n \lambda_{H_1}}{2} \left\| t h_{11}^{1(i)} \right\|_2^2.$$

This problem is strongly convex, quadratic and simple enough to give the following: if $\alpha > 1$, then $\left\| h_{11}^{1(1)} \right\|_2^2 > \left\| h_{11}^{1(2)} \right\|_2^2$ and simultaneously $\| W_L x - e_1 \|_2^2 > \| W_L y - e_1 \|_2^2$. This means that $H_1^{(2)}$ is a strictly better solution than $H_1^{(1)}$ – a contradiction.

**Step 3: Contradiction by zero gradient condition.** By optimality of both solutions we get

$$\frac{\partial \mathcal{L}}{\partial W_L} \bigg|_{(H_1, W_1, \ldots, W_L) = (H_1^{(1)}, W_1^*, \ldots, W_L^*)} = 0 = \frac{\partial \mathcal{L}}{\partial W_L} \bigg|_{(H_1, W_1, \ldots, W_L) = (H_1^{(2)}, W_1^*, \ldots, W_L^*)}.$$

An application of the chain rule gives

$$\frac{\partial \mathcal{L}}{\partial W_L} = \frac{\partial \mathcal{L}_F}{\partial \tilde{H}_{L+1}} \frac{\partial \tilde{H}_{L+1}}{\partial W_L} + \lambda_{W_L} W_L = \frac{\partial \mathcal{L}_F}{\partial \tilde{H}_{L+1}} H_L^T + \lambda_{W_L} W_L,$$

where $\tilde{H}_{L+1}$ is the output of our model. Plugging this back to the previous equation and using that $W_L^*$ is the same in both expressions, we get

$$\frac{\partial \mathcal{L}_F}{\partial \tilde{H}_{L+1}} \bigg|_{(H_1, W_1, \ldots, W_L) = (H_1^{(1)}, W_1^*, \ldots, W_L^*)} (H_L^{(1)})^T = \frac{\partial \mathcal{L}_F}{\partial \tilde{H}_{L+1}} \bigg|_{(H_1, W_1, \ldots, W_L) = (H_1^{(2)}, W_1^*, \ldots, W_L^*)} (H_L^{(2)})^T.$$

Let us denote

$$A = \frac{\partial \mathcal{L}_F}{\partial \tilde{H}_{L+1}} \bigg|_{(H_1, W_1, \ldots, W_L) = (H_1^{(1)}, W_1^*, \ldots, W_L^*)}, \quad B = \frac{\partial \mathcal{L}_F}{\partial \tilde{H}_{L+1}} \bigg|_{(H_1, W_1, \ldots, W_L) = (H_1^{(2)}, W_1^*, \ldots, W_L^*)}.$$

Due to the separability of $\mathcal{L}_F$ with respect to the columns of $H_1$ (and, thus, also of $H_l, \tilde{H}_l$ for all $l \leq L + 1$), we get that the matrices $A, B$ can only differ in their first columns and are identical otherwise. We denote these columns $a, b$ for $A, B$, respectively. This implies that $ax^T = by^T$.

Now we exclude a few cases. First, neither $a$ nor $b$ can be zero, because by the exact formula that exists for them, this would mean that exact fit was achieved for either of the columns. This is impossible with non-zero weight-decay, because decreasing the norm of the column of $H_1^{(1)}$ or $H_1^{(2)}$ that achieves the exact fit by a sufficiently small value would necessarily lead to an improvement on the objective value. Moreover, if $x = y = 0$, then this is a contradiction with the assumption that $x \neq y$. Finally, the case $x \neq y = 0$ or $0 = x \neq y$ is excluded already in the *step 2*.

Thus we get $x, y, a, b$ are all non-zero. Looking at any fixed row (column) of $ax^T$ and $by^T$ we see that necessarily $x, y$ $(a, b)$ are aligned. However, this case is already solved in *step 2* and leads to $x = y$, which is the contradiction. This concludes the proof. $\qquad\square$

**Theorem 8.** *Denote by L-DUFM$_\epsilon$ the equivalent of (1), with $\sigma$ replaced by $\sigma_\epsilon$. Let $D = \max\{d_2, d_3, \ldots, d_L\}$ and $\bar\lambda = \lambda_{H_1}\lambda_{W_1}\ldots\lambda_{W_L}$, with the regularization parameters upper bounded by $1/(L+1)$. Then, for any globally optimal solution of the L-DUFM$_\epsilon$ problem, the distance between any two feature vectors of the same class in any layer is at most*

$$\frac{6\epsilon\sqrt{D(L+1)}}{(L+1)^{L+1}\bar\lambda\sqrt{n}}. \tag{5}$$

*Proof.* In order not to mix approximation technical details with the gist of the proof, we split the argument into two parts: in the first part, we provide a heuristic for $\epsilon = 0$ by assuming that ReLU is differentiable at 0; in the second part, we discuss what changes in the proof if we use the relaxation and then execute a technical computation that bounds the error based on the strength of the approximation (the size of $\epsilon$). It is useful to split the loss function $\mathcal{L}$ into the two terms $\mathcal{L}_F$ and $\mathcal{L}_R$. The former represents the fit part of the loss (which penalizes for deviation of the predictions from $Y$), and the latter represents the regularization part of the loss.

**Part 1: Heuristic for $\epsilon = 0$.** We start identically to the proof of Theorem 6. As in *Step 1* of that argument, if we have a globally optimal solution that does not exhibit DNC1 in the first layer, we can construct two different solutions of the $n = 1$ L-DUFM problem where the two solutions only differ in the first column of the $H_1$ matrix. Let us denote two such constructions $H_1^x \neq H_1^y$, where we denote $x \neq y$, to be the mentioned first columns, respectively. We emphasize that $H_1^x$, $H_1^y$ both form optimal solutions of the $n = 1$ L-DUFM with *the same* tuple of weight matrices $(W_1^*, \ldots, W_L^*)$. In particular, this means that

$$\left.\frac{\partial\mathcal{L}}{\partial W_1}\right|_{(H_1,W_1,\ldots,W_L)=(H_1^x,W_1^*,\ldots,W_L^*)} = 0 = \left.\frac{\partial\mathcal{L}}{\partial W_1}\right|_{(H_1,W_1,\ldots,W_L)=(H_1^y,W_1^*,\ldots,W_L^*)}.$$

An application of the chain rule gives

$$\frac{\partial\mathcal{L}}{\partial W_1} = \frac{\partial\mathcal{L}_F}{\partial\tilde{H}_2}\frac{\partial\tilde{H}_2}{\partial W_1} + \lambda_{W_1}W_1 = \frac{\partial\mathcal{L}_F}{\partial\tilde{H}_2}H_1^T + \lambda_{W_1}W_1.$$

Plugging this back to the previous equation and using that the $W_1^*$ is the same in both expressions, we get

$$\left.\frac{\partial\mathcal{L}_F}{\partial\tilde{H}_2}\right|_{(H_1,W_1,\ldots,W_L)=(H_1^x,W_1^*,\ldots,W_L^*)}(H_1^x)^T = \left.\frac{\partial\mathcal{L}_F}{\partial\tilde{H}_2}\right|_{(H_1,W_1,\ldots,W_L)=(H_1^y,W_1^*,\ldots,W_L^*)}(H_1^y)^T.$$

Let us denote

$$A = \left.\frac{\partial\mathcal{L}_F}{\partial\tilde{H}_2}\right|_{(H_1,W_1,\ldots,W_L)=(H_1^x,W_1^*,\ldots,W_L^*)}, \quad B = \left.\frac{\partial\mathcal{L}_F}{\partial\tilde{H}_2}\right|_{(H_1,W_1,\ldots,W_L)=(H_1^y,W_1^*,\ldots,W_L^*)}.$$

Due to the separability of $\mathcal{L}_F$ with respect to the columns of $H_1$ (and, thus, also $H_l$ for all $l \leq L$), we get that the matrices $A, B$ can only differ in their first columns and are identical otherwise. We denote these columns $a, b$ for $A, B$, respectively. This implies that $ax^T = by^T$.

Now, we treat a few cases. First, assume $a = b = 0$. Since it holds that $(H_1^x, W_1^*, \ldots, W_L^*)$ is the optimal solution, then necessarily

$$0 = \left.\frac{\partial\mathcal{L}}{\partial h_{11}^1}\right|_{(H_1,W_1,\ldots,W_L)=(H_1^x,W_1^*,\ldots,W_L^*)} = (W_1^*)^T a + \lambda_{H_1}x \iff 0 = x.$$

Similarly, we get $y = 0$, but that is a contradiction with $x \neq y$. Therefore, at least one of $a, b$ is non-zero. Next assume $x = 0$. Then $ax^T = 0$. If $b = 0$ then $y = 0$ and we have a contradiction. On the other hand, if $b \neq 0$, then the row of $by^T$ that corresponds to a non-zero entry of $b$ must be zero and thus $y = 0$. Similarly, if $y = 0$ we can get $x = 0$.

Let us therefore assume $x, y$ are both non-zero, which also implies $a, b$ are both non-zero. Looking at any fixed row (column) of $ax^T$ and $by^T$ we see that necessarily $x, y$ $(a, b)$ are aligned. Let us write $x = \alpha y$ and $\alpha a = b$ for some $\alpha \neq 0$. We will first show that if $\alpha > 0$ then necessarily $\alpha = 1$ and $x = y$. For this, let us fix any ray $r \in \mathbb{R}^{d_1}$. The ray $r$ represents a set of possible first columns in

$H_1$. Let us fix any $(W_1, \ldots, W_L)$. Since $\mathcal{L}$ is separable in the columns of $H_1$, we can consider an optimization over $\beta \geq 0$ to minimize $\mathcal{L}$ on $(\beta r, W_1, \ldots, W_L)$. However, $\mathcal{L}_F$ is convex by assumption and the mapping $h_{11}^1 \to h_{11}^L$ is ray-linear, therefore $\mathcal{L}_F$ is convex in $\beta$. Moreover $\mathcal{L}_R$ is strongly convex in $h_{11}^1$ and therefore $\mathcal{L}$ is strongly convex in $\beta$. This means it has a unique optimal solution $\beta^*$.

We have just showed that, if $\alpha > 0$, then since $x, y$ are aligned and thus lie on the same ray and are both optimal together with the same tuple of weight matrices, they must necessarily be identical and so $\alpha = 1$, $x = y$. This is a contradiction and thus we are left with the case $\alpha < 0$. Denote $s = W_1^* x$, $t = W_1^* y$. From linearity, we have $s = \alpha t$. Note that if $s = t = 0$, then necessarily $x = y = 0$ because they don't have any effect on $\mathcal{L}_F$ and they minimize $\mathcal{L}$ at 0. Thus, $s, t$ are non-zero and negative entries of $s$ are positive entries of $t$ and vice-versa. This means that $\sigma(s)$ and $\sigma(t)$ have different sets of positive entries. However, from the optimality of the both solutions we know:

$$\left. \frac{\partial \mathcal{L}}{\partial W_2} \right|_{(H_1, W_1, \ldots, W_L) = (H_1^x, W_1^*, \ldots, W_L^*)} = 0 = \left. \frac{\partial \mathcal{L}}{\partial W_2} \right|_{(H_1, W_1, \ldots, W_L) = (H_1^y, W_1^*, \ldots, W_L^*)}.$$

Again, using the chain rule we get:

$$\frac{\partial \mathcal{L}}{\partial W_2} = \frac{\partial \mathcal{L}_F}{\partial \tilde{H}_3} \frac{\partial \tilde{H}_3}{\partial W_2} + \lambda_{W_2} W_2 = \frac{\partial \mathcal{L}_F}{\partial \tilde{H}_3} H_2^T + \lambda_{W_2} W_2.$$

Plugging this back to the previous equation and using that $W_2^*$ is the same in both expressions, we get

$$\left. \frac{\partial \mathcal{L}_F}{\partial \tilde{H}_3} \right|_{(H_1, W_1, \ldots, W_L) = (H_1^x, W_1^*, \ldots, W_L^*)} (H_2^x)^T = \left. \frac{\partial \mathcal{L}_F}{\partial \tilde{H}_3} \right|_{(H_1, W_1, \ldots, W_L) = (H_1^y, W_1^*, \ldots, W_L^*)} (H_2^y)^T.$$

Let us denote

$$C = \left. \frac{\partial \mathcal{L}_F}{\partial \tilde{H}_3} \right|_{(H_1, W_1, \ldots, W_L) = (H_1^x, W_1^*, \ldots, W_L^*)} \quad , \quad D = \left. \frac{\partial \mathcal{L}_F}{\partial \tilde{H}_3} \right|_{(H_1, W_1, \ldots, W_L) = (H_1^y, W_1^*, \ldots, W_L^*)}.$$

Due to the separability of $\mathcal{L}_F$ with respect to the columns of $H_1$ (and, thus, also $H_l$ for all $l \leq L$), we get that the matrices $C, D$ can only differ in their first columns and are identical otherwise. We denote these columns $c, d$ for $C, D$ respectively. This implies that $c\sigma(s)^T = d\sigma(t)^T$.

As above, if either $c$ or $d$ is zero, using the chain rule we would get that $x$ or $y$ (respectively) are zero too, which cannot happen. Therefore, both $c$ and $d$ are non-zero and $\sigma(s), \sigma(t)$ must be aligned. Since they are non-zero, non-negative and with different supports, we have reached a contradiction. This proves that $\alpha < 0$ is also impossible and the only possible case is $\alpha = 1$, but that forces $x = y$, which is also a contradiction.

**Part 2: Relaxation to ReLU$_\epsilon$.** The difficulty in making the previous heuristic rigorous is that ReLU is not differentiable at 0 and, thus, we do not have the desired analytical statement that global solutions must necessarily admit zero derivative (because the loss might not be differentiable at them at all). If we tried to use the same proof as in *part 1* with the differentiable relaxed version of ReLU – ReLU$_\epsilon$, an issue would occur when showing that, if $x, y$ are aligned and $\alpha \geq 0$, then $\alpha$ must be 1. The reason is that the mapping $h_{11}^1 \to h_{11}^L$ is no longer ray-linear and the corresponding optimization problem on any fixed ray is no longer quadratic and strongly convex. However, we can prove that the optimization problem admits a solution that is *close* to the solution of the corresponding ReLU optimization problem. For this, let us fix any direction $r$ such that $\|r\| = 1$ and an optimization parameter $t \geq 0$, and define the following two losses:

$$L_0(t, r) = \frac{1}{2K} \left\| W_L^* \sigma(W_{L-1}^* \sigma(\ldots W_2^* \sigma(W_1^* tr) \ldots)) - e_1 \right\|_F^2 + \frac{n\lambda_{H_1}}{2} t^2, \tag{19}$$

$$L_\epsilon(t, r) = \frac{1}{2K} \left\| W_L^* \sigma_\epsilon(W_{L-1}^* \sigma_\epsilon(\ldots W_2^* \sigma_\epsilon(W_1^* tr) \ldots)) - e_1 \right\|_F^2 + \frac{n\lambda_{H_1}}{2} t^2. \tag{20}$$

We now bound $\max_{t \geq 0} |L_0(t, r) - L_\epsilon(t, r)|$. For this, we fix any $t \geq 0$, and bound the accumulated $l_2$ error of using $\sigma_\epsilon$ instead of $\sigma$ throughout the layers. For this, a useful statement that we will need is the following:

$$\frac{\lambda_{W_L}}{2} \|W_L^*\|_F^2 = \frac{\lambda_{W_{L-1}}}{2} \|W_{L-1}^*\|_F^2 = \cdots = \frac{\lambda_{W_1}}{2} \|W_1^*\|_F^2 = \frac{n\lambda_{H_1}}{2} \|H_1^*\|_F^2 \leq \frac{1}{2(L+1)}.$$

This is true because, by a simple computation, we get that the terms must be balanced and the inequality comes from the fact that the solution $(0, 0, \ldots, 0)$ achieves full loss $1/2$ in the $L$-DUFM as well as $L$-DUFM$_\epsilon$ problems and, thus, $\mathcal{L}_R$ is trivially upper-bounded by this. This implies that, for each $W_l$ (and $H_1$), $\|W_l\| \leq \|W_l\|_F \leq \frac{1}{\sqrt{(L+1)\lambda_{W_l}}}$.

For ease of exposition, we set $t = 1$ (the same argument would work for all $t$). We see that $W_1 r$ does not introduce any $l_2$ error. Then, the $l_2$ error that is introduced in the application of the ReLU is trivially upper-bounded by $\epsilon\sqrt{d_2}$. After applying $W_2$, we can use the bound on the operator norm $\|W_2\|$ obtained above to upper bound the propagated $l_2$ error by

$$\frac{1}{\sqrt{(L+1)\lambda_{W_2}}}\epsilon\sqrt{d_2}.$$

After that, we obtain an additive $l_2$ error of $\epsilon\sqrt{d_3}$ by applying $\sigma_\epsilon$, which gives a total $l_2$ error of

$$\frac{1}{\sqrt{(L+1)\lambda_{W_2}}}\epsilon\sqrt{d_2} + \epsilon\sqrt{d_3}.$$

Then, again, we multiply this whole expression with $\frac{1}{\sqrt{(L+1)\lambda_{W_3}}}$ to account for the multiplication by $W_3$. Inductively, the upper bound on the $l_2$ error in the output space is

$$\epsilon\sum_{l=2}^{L}\frac{\sqrt{d_l}}{(L+1)^{\frac{L-l+1}{2}}}\prod_{j=l}^{L}\frac{1}{\sqrt{\lambda_{W_j}}}.$$

This can be further upper bounded as

$$\frac{\epsilon\sqrt{D}\sqrt{\lambda_{H_1}\lambda_{W_1}}}{(L+1)^{\frac{L-1}{2}}\sqrt{\overline{\lambda}}}.$$

Using the triangle inequality, the upper bound on $|L_0(t, r) - L_\epsilon(t, r)|$ is this expression squared. On the other hand, the second derivative of $L_0(t, r)$ with respect to $t$ is lower bounded by $n\lambda_{H_1}$.

Given two functions $f_1$ and $f_2$, with $f_1$ strongly convex with second derivative at least $c$ and $f_2$ everywhere at most $d$ distant from $f_1$, then the distance between their global minimizers is at most $2\sqrt{d/c}$. Applying this to our case, we get that the distance between the minimizers $t_0$ and $t_\epsilon$ of $L_0(t, r)$ and $L_\epsilon(t, r)$ is at most

$$\frac{2\epsilon\sqrt{D(L+1)}}{(L+1)^{\frac{L+1}{2}}\sqrt{\lambda n}}.$$

Since the ray $r$ is unit-norm, this is also the upper bound on the distance between two feature vectors of any globally optimal solution in the first layer. To obtain the upper bound on the distance between two vectors in any layer, we proceed as follows.

Assume we have two input vectors of the first class (now we know they need to be aligned): $x^1, y^1 = \alpha^1 x^1$, where $\alpha^1 > 1$. As before, $\tilde{x}^l, x^l, \tilde{y}^l, y^l$ are the $l$-th layer representations of these vectors before and after $\sigma_\epsilon$. If we would compute $\frac{\partial\mathcal{L}}{\partial W_l}$ with respect to any layer $l \geq 2$ and used the same arguments as in part 1 (that still carry to the analysis for $\sigma_\epsilon$), we would find out that in each layer, we need to have $y^l = \alpha^l x^l$. Moreover, we can assume that $\tilde{x}^l, x^l, \tilde{y}^l, y^l$ are non-zero at all layers because otherwise the same argument in *part 1* would trivialize the rest of the proof. By a simple inductive argument, since $\alpha_1 > 1$, we know that $\alpha_2 > 1$ as well because $\sigma_\epsilon$ is strictly increasing on $[0, \infty)$. Similarly, $\alpha_l > 1$ for all $l$. Note that, if there exists at least one index $i$ on which $x_i^l$ is bigger or equal than $\epsilon$, then necessarily $\alpha^l = \alpha^{l-1}$, because $\sigma_\epsilon$ is the identity on inputs of at least $\epsilon$, and the $\alpha_l$ can be uniquely determined from $y_i^l/x_i^l = y_i^{l-1}/x_i^{l-1}$. Having $\alpha^l = \alpha^{l-1}$ makes us have more control over the distance between $x^l$ and $y^l$. If this fails to hold at some layer, we need a separate analysis.

For this, let $l_0$ denote the first layer (assuming it exists), where $\tilde{x}^{l_0}$ does not have any entries that are bigger or equal than $\epsilon$. This means that $\|x^{l_0}\| \leq \epsilon\sqrt{d_{l_0}}$. We now compute the maximal possible norm of $y^{l_0}$. Since $\alpha_1 = \alpha_2 = \cdots = \alpha_{l_0-1}$, for all $2 \leq l < l_0$ and for each index $i$, either $\tilde{x}_i^l, \tilde{y}_i^l \leq \epsilon$ or $\epsilon \leq \tilde{x}_i^l, \tilde{y}_i^l$. Otherwise, since $\sigma_\epsilon(x) < x \; \forall x \in (0, \epsilon)$, $y_i^l/x_i^l > \alpha_l$, a contradiction. Thus, we get

$\left\|\tilde{x}^l - \tilde{y}^l\right\| \geq \left\|x^l - y^l\right\|$. Therefore, $\left\|x^l - y^l\right\|$ can only grow by applying $W_l$. Thus, after applying $W_1, W_2, \ldots, W_{l_0-1}$ and using the operator norm bound with all of the matrices, we get

$$\left\|x^l - y^l\right\| \leq \frac{2\epsilon\sqrt{D(L+1)}}{(L+1)^{\frac{L+1}{2}}\sqrt{\bar{\lambda}n}} \frac{1}{(L+1)^{\frac{l_0-1}{2}}\sqrt{\lambda_{W_1}\lambda_{W_2}\ldots\lambda_{W_{l_0-1}}}}.$$

Combining with $\left\|x^{l_0}\right\| \leq \epsilon\sqrt{d_{l_0}}$ and using the triangle inequality, we get

$$\left\|y^l\right\| \leq \frac{2\epsilon\sqrt{D(L+1)}}{(L+1)^{\frac{L+1}{2}}\sqrt{\bar{\lambda}n}} \frac{1}{(L+1)^{\frac{l_0-1}{2}}\sqrt{\lambda_{W_1}\lambda_{W_2}\ldots\lambda_{W_{l_0-1}}}} + \epsilon\sqrt{d_{l_0}}.$$

Applying the operator norm bound with $W_{l_0}, W_{l_0+1}, \ldots, W_{L-1}$, we get:

$$\begin{aligned}
\left\|y^L\right\| &\leq \frac{2\epsilon\sqrt{D(L+1)}}{(L+1)^{\frac{L+1}{2}}\sqrt{\bar{\lambda}n}} \frac{1}{(L+1)^{\frac{L-1}{2}}\sqrt{\lambda_{W_1}\lambda_{W_2}\ldots\lambda_{W_{L-1}}}} \\
&\quad + \epsilon\sqrt{d_{l_0}} \frac{1}{(L+1)^{\frac{L-l_0}{2}}\sqrt{\lambda_{W_{l_0}}\lambda_{W_{l_0+1}}\ldots\lambda_{W_{L-1}}}} \\
&\leq \frac{3\epsilon\sqrt{D(L+1)}}{(L+1)^{L+1}\bar{\lambda}\sqrt{n}}.
\end{aligned}$$

Since $\left\|x^L\right\| \leq \left\|y^L\right\|$ we can use the triangle inequality again to obtain the final:

$$\left\|x^L - y^L\right\| \leq \frac{6\epsilon\sqrt{D(L+1)}}{(L+1)^{L+1}\bar{\lambda}\sqrt{n}}.$$

Note that the same bound is an upper bound for any layer, since in each layer we are increasing the upper bound by at least a factor $(L+1)^{-1/2}\lambda_{W_l}^{-1/2}$.

In case such a layer $l_0$ does not exist, the upper bound above is still valid, since we can simply use the error propagation through all the layers, as we did for $l < l_0$ in the previous case. We would arrive at the same bound as above but with 4 instead of 6 as a multiplicative factor (because we would not introduce the additive error $\epsilon\sqrt{d_{l_0}}$). $\qquad\square$

# B   Further experimental evidence

## B.1   $L$-DUFM experiments

**Experiments on small number of classes and layers.**   For $L = 3$, we weren't able to find any solutions that would outperform DNC for $K \leq 6$. For $K = 7$, the difference in the losses is extremely small. For $K = 6$, we did find *some* low rank solutions, but these were either slightly worse than DNC or the losses were the same up to four decimal places. Already for $L \geq 4$, we find low-rank solutions for all $K \geq 3$. We present some of them for $K \in \{3, 4, 5\}$ in Figure 5. Note, that all these solutions were found *automatically* by gradient descent through the optimization process.

We specifically highlight the solution for $K = 4$, as it represents another graph structure different from SRG. Namely, this solution is based on the square graph. We see that each column (edge) has positive scalar product with exactly two other columns (the corresponding edges share a vertex), and it is orthogonal to one other column (the non-touching sides of a square). Each row (vertex) has only two positive values, and these correspond to the two sides (edges) to which they belong.

**Experiments on large number of classes or layers.**   For large $L$ or $K$, the behavior of the low-rank solutions is slightly different from the behavior when both $L$ and $K$ are moderate. When $l$ is close to $L$, it no longer holds that $\tilde{M}_l = M_l$, and the rank of $M_l$ is larger as the singular values do not decay sharply to 0. However, we note that the singular values are rather small and, thus, it is not clear whether they would disappear with a longer training (we trained for 200000 steps of full gradient descent with learning rate 0.5). In Figure 6, we take $L = 7, K = 15$, and weight decay 0.0025 in all layers. We highlight that $\tilde{M}_6 \neq M_6$ and the singular values of $M_6$ are small but non-zero after the 3 dominant values. We also highlight that the rank of previous layers is 3, which is remarkably small (and also smaller than the rank of the SRG solution).

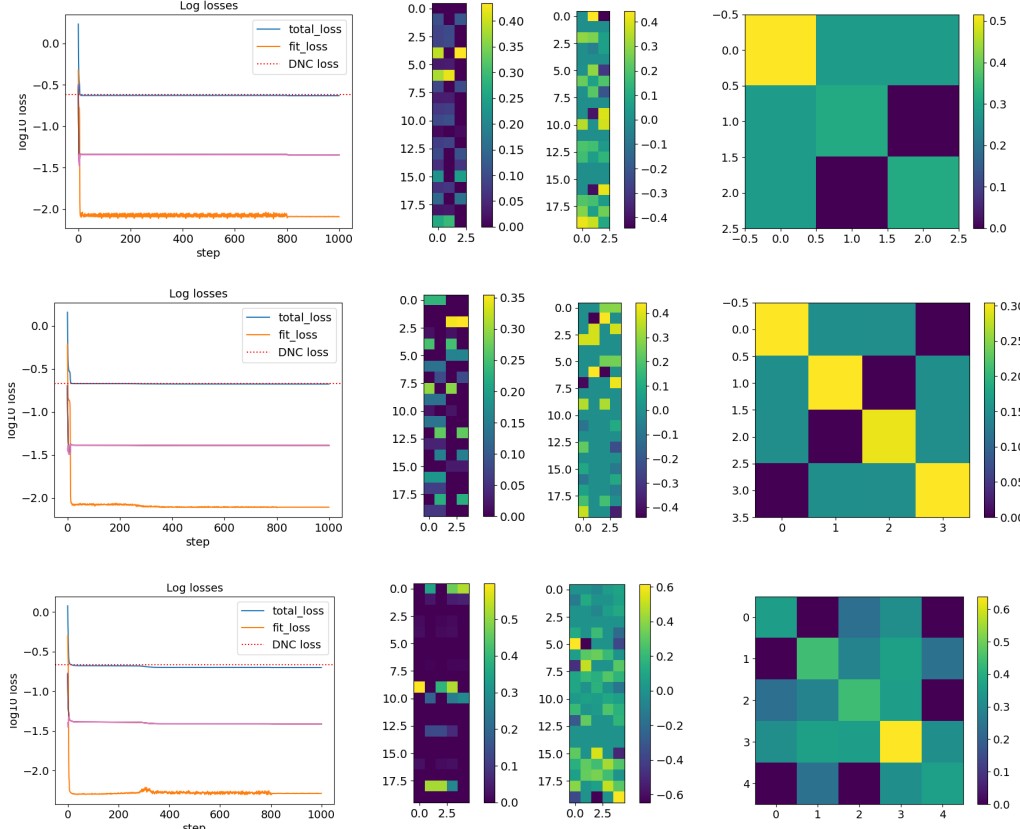

Figure 5: 4-DUFM training for $K = 3$ (**top**), $K = 4$ (**middle**), and $K = 5$ (**bottom**). **Left:** Loss progression, also decomposed into the fit and regularization terms. **Middle left:** Visualization of the matrix $M_3$. **Middle right:** Visualization of the matrix $\tilde{M}_4$. **Right:** Visualization of the matrix $M_3^T M_3$.

**SRG solutions.** In Figure 1 presented in the main body, we recover the SRG solution for $K = 10$. In Figures 7 and 8, we show that solutions very similar to SRG are recovered for $K = 6$ and $K = 15$, respectively. The only difference with SRG is in the construction of $\tilde{M}_L$. We note that the losses of these solutions are slightly lower than the loss of our construction, which proves that the SRG solution itself is not necessarily globally optimal.

## B.2 End-to-end experiments with DUFM-like regularization

We complement the experiments of Figure 3 with two extra ablation studies for ResNet20 trained on CIFAR-10 with a 4-layer MLP head. We focus on the dependence of the average rank on the weight decay and the learning rate, and present the results in Figure 9. The weight decay has a clear effect on the rank of the solutions found by gradient descent, similarly to the results in Figure 3 for the $L$-DUFM model. The effect of the learning rate is slightly less clear, but we still see a general downward trend.

## B.3 End-to-end experiments with standard regularization

Finally, in Figure 10, we include the analysis of the average rank as a function of weight decay in the standard regularization setting for training on the MNIST dataset. The results confirm the trend from the previous experimental settings, showing that the weight decay strength is a crucial predictor of the final rank even in standard regularization setting, which has a different loss landscape compared to the $L$-DUFM.

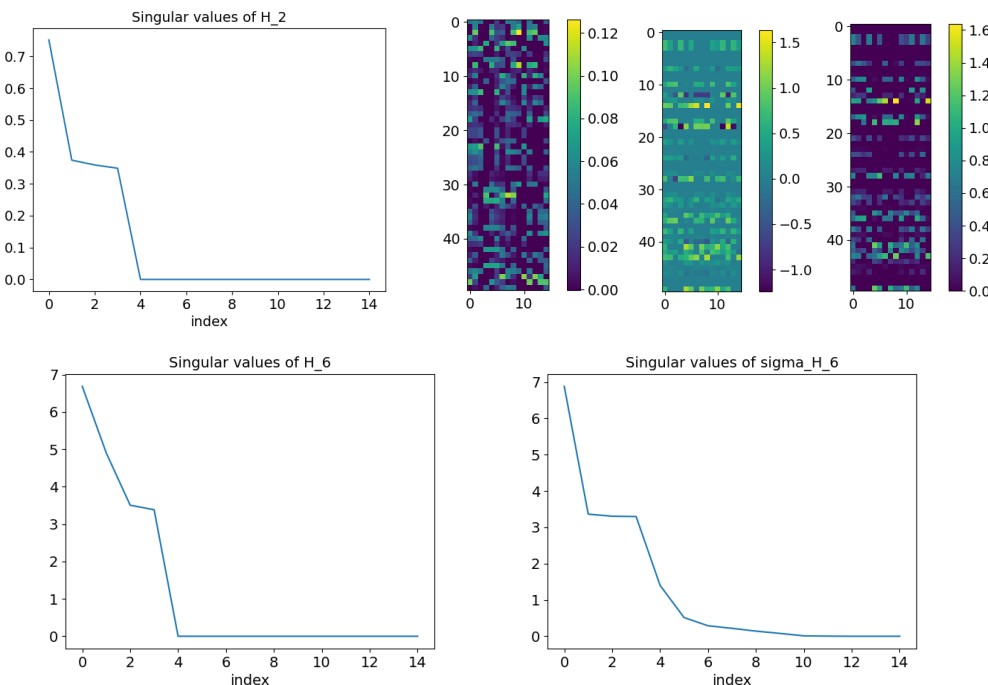

Figure 6: Class-mean matrices and singular values at convergence for a DUFM model with $K = 15$ and $L = 7$. **Top row:** Singular values of $\tilde{M}_2$, and visualization of the matrices $\tilde{M}_2, \tilde{M}_6, M_6$ and **Bottom row:** Singular values of $\tilde{M}_6$ and $M_6$.

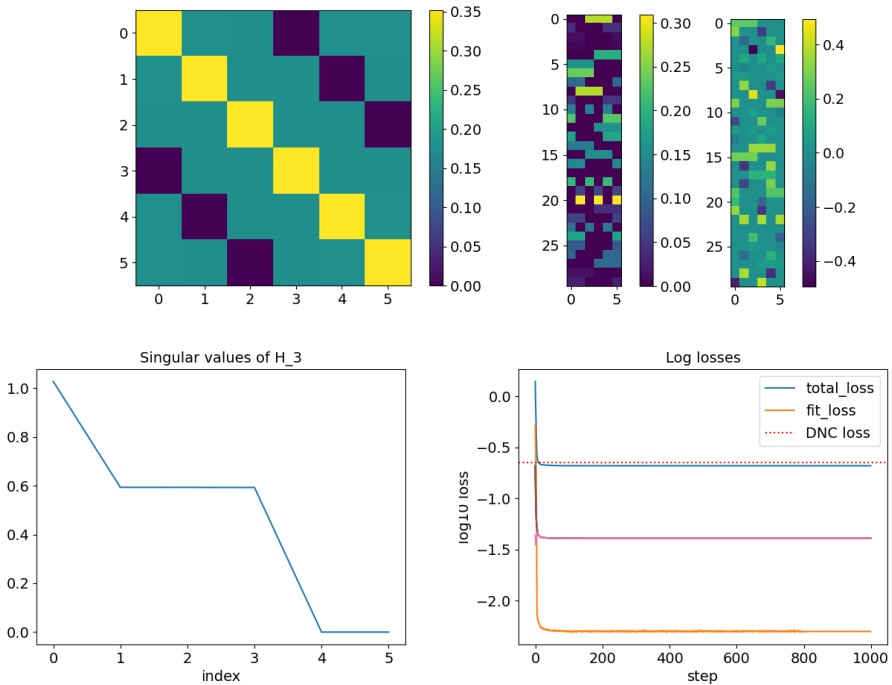

Figure 7: 4-DUFM training for $K = 6$. **Top row:** Visualization of the matrices $\tilde{M}_3^T \tilde{M}_3, \tilde{M}_3$, and $\tilde{M}_4$. **Bottom row:** Singular values of $H_3$, and loss progression including its decomposition into fit and regularization terms.

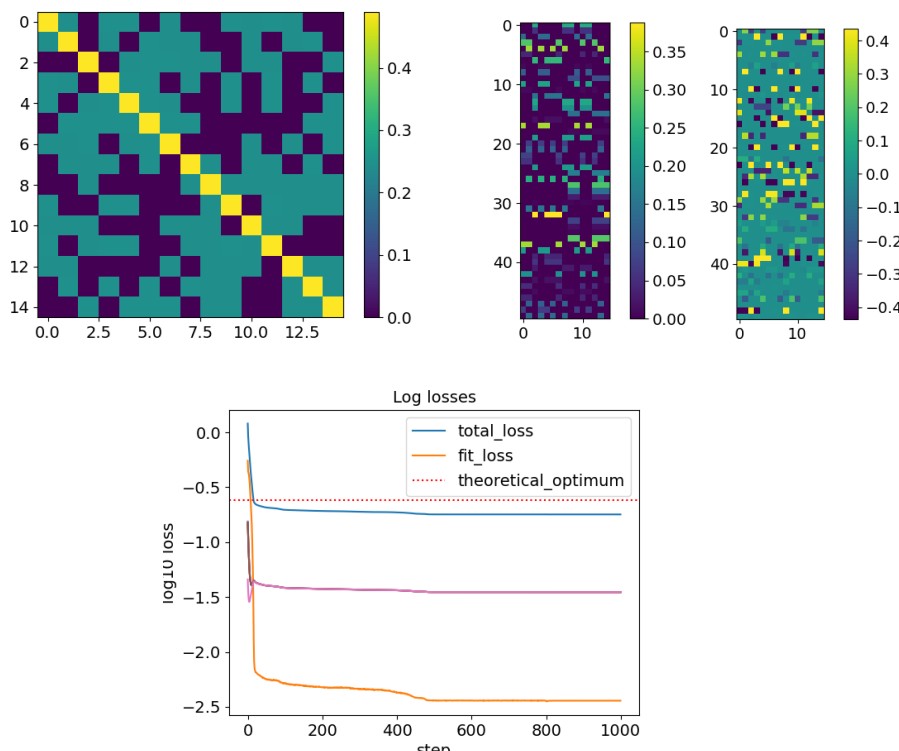

Figure 8: 4-DUFM training for $K = 15$. **Top row:** Visualization of the matrices $\tilde{M}_3^T \tilde{M}_3, \tilde{M}_3$ and $\tilde{M}_4$. **Bottom row:** Loss progression including its decomposition into fit and regularization terms.

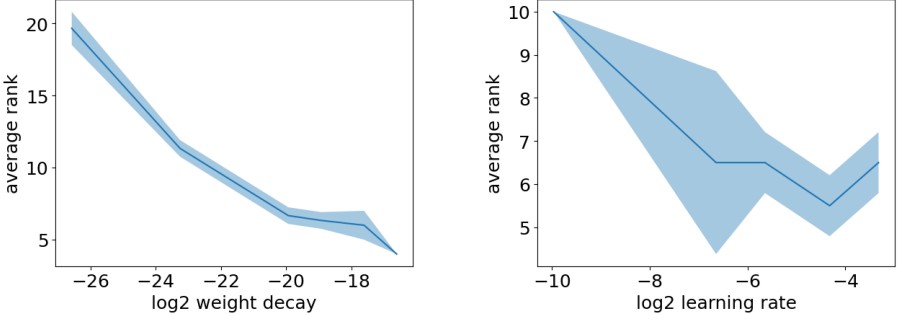

Figure 9: Average ranks as a function of $\log_2$ weight decay (**left**) and $\log_2$ learning rate (**right**). We trained ResNet20 with 4-layer MLP head on CIFAR-10. The experiments are averaged over three and two independent runs, respectively.

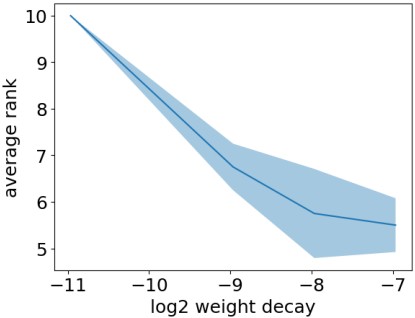

Figure 10: Average rank as a function of the $\log_2$ weight decay. We trained ResNet20 with 5-layer MLP head on MNIST. The experiments are averaged over 4 independent runs.

