# OpenReview forum: "Neural collapse vs. low-rank bias: Is deep neural collapse really optimal?"
_NeurIPS.cc/2024/Conference — NeurIPS 2024 poster_

### Official Review · Reviewer_hQr6 · 2024-07-08

**Soundness:** 2
**Presentation:** 2
**Contribution:** 3
**Rating:** 5
**Confidence:** 4

**Summary:**

The paper studies the optimality of neural collapse in intermediate layers of a deep neural network on classification tasks with more than 2 classes.

1. The paper shows that the NC2 property (formation of a simplex ETF/orthogonal frame) in the intermediate layer features is not optimal, and constructs a strongly regular graph (SRG) based lower-rank solution leading to lower loss than previously proposed deep neural collapse (DNC) optimal solutions.

2. The NC1 property (variability collapse of features) is still shown to be optimal in these settings.

**Strengths:**

The key contribution of this paper is the construction of a strongly regular graph (SRG) based lower-rank solution for a generic $L > 2$ layer neural network trained on classification tasks with $K > 2$ classes. The authors show that this construction leads to lower training loss than the deep neural collapse-based solution.

**Weaknesses:**

1. There seems to be an error in the Sherman-Morrison formula-based inverse in the final result of Lemma 12, i.e, $(I + \frac{1}{r-1}11^\top)^{-1} = I - \frac{1}{2(r-1)}11^\top$. (The authors can correct me on this). Since this result is used in Lemma 13 and affects the results in Theorem 5, verifying this result is important.

2. Can Definition 4 be split into multiple parts for better readability? For instance, split it into parts w.r.t:

- collapsed class means are unaffected by non-linearity.
- The class means are scaled versions of $T_r$ where each row is scaled individually.
- Choice of $A_L$ for the final layer.

This breakdown highlights a major assumption that the non-linearity is not in effect till the last layer. Additionally, the complete set of trainable and non-trainable parameters should be made clear. The full definition in Appendix A.1 mentions extra trainable terms corresponding to $q$ terms.

3. In section 6.1 (DUFM training) Figure 2 (first row, last column), the plots for the singular values of class means seem to have a rank of around 7 or 8 for intermediate layers as well. The ideal rank $K=10$ might not exactly hold for intermediate layers but these observations should be discussed since it is one of the main claims in the paper.

4. The experimental results in section 6.3 for MNIST datasets claim that the network trained using gradient descent is the SRG solution. A justification for this observation is missing. The heatmap for MNIST in Figure 4 corresponds to $M_5$ which as per notation indicates the class-means of the penultimate layer post-activations. However, shouldn't the SRG solution correspond to similar heat maps across intermediate layers? Also, a statement such as low-rank implying an SRG solution is not justified.

**Questions:**

1. Do numerical experiments with the 5-layer MLP head include a ReLU activation for L-DUFM and end-end training?

2. The authors mention in section 6.3 that "We also observe that the rank deficiency is the strongest in the mid-layer of the MLP head". Can further justification be provided for this? Especially which aspect of the spectrum of Layer 3 or 4 in Figure 5 justifies this observation?

3. Since the SRG solution relies on feature class means being non-negative, how valid is this assumption (based on your practical experiments)?

nit: In the notation for $H_L = \sigma(W_{L-1}X_{L-1})$ in section 3, shouldn't $X_{L-1}$ be replaced with $H_{L-1}$ ? The same applies to $\widetilde{X}_l$.

nit: Please fix the formula for NC1 in the first line of section 6. The pseudo-inverse is not required when dividing the traces.

---

> ### Author Rebuttal · Authors · 2024-08-02
>
> We thank the reviewer for a detailed review. The typos pointed out by the reviewer will be corrected in the revision, and we thank the reviewer for spotting those. We address all the other concerns below (including the technical issue about the Sherman-Morrison formula), and we would like to kindly ask the reviewer to raise the score (together with the soundness score) in case no additional issue remains.
>
> **An issue with Sherman-Morrison formula.**
>
> We carefully checked our computation and believe there is no error. The matrix we have to invert is indeed $I+\frac{1}{r-2}\mathbb{1}\mathbb{1}^T$ (not $I+\frac{1}{r-1}\mathbb{1}\mathbb{1}^T$, as in the review), as this is the main factor of the previous computation of $T_r T_r^T$, where we exchange the multiplier of $\mathbb{1}\mathbb{1}^T$ from $\frac{1}{r-1}$ to $\frac{1}{r-2}$ by pulling $\frac{r-2}{r-1}$ in front of the parenthesis. Then, we use the Sherman-Morrison formula $(A+uv^T)^{-1}=A^{-1}-\frac{A^{-1}uv^TA^{-1}}{1+u^TA^{-1}v}$, where $A$ is the identity matrix and both $u$ and $v$ are $\frac{1}{\sqrt{r-2}}\mathbb{1}$. Thus, we get $I$ as the left summand, $\frac{1}{r-2}\mathbb{1}\mathbb{1}^T$ as the numerator in the right summand, and $1+\frac{r}{r-2}=\frac{2(r-1)}{r-2}$ as the denominator. The term $r-2$ cancels in numerator and denominator, and we are left with the right hand side we write in our computation.
>
> **The definition 4 should be split and clarified.**
>
>  Thank you for pointing this out, we agree that this definition is rather complicated. In the revision, we will split the definition into different bullet points where we distinguish the construction of the intermediate layers and the final layer, highlight the non-negativity of intermediate activations and more clearly define the trainable parameters which are the scales of the involved matrices.
>
> **In Figure 2 the rank is 7-8, which is not K=10. This should be discussed more.**
>
>  The rank 5-8 we found in our experiments is indeed smaller than the rank of DNC solution, which in this case is 10. This is exactly in agreement with Theorem 5, which shows that NC2 solutions are not globally optimal. We will make this explicit clarification in the revision, adding to the discussion in lines 259-263.
>
> **The sentence “solution found by gradient descent is the SRG solution” from line 313 should be more justified.**
>
> Thank you for pointing this out, we will clarify the sentence in the revision. What we meant is that the particular solution (i.e. a single run) whose gram matrix is displayed in lower row right of Figure 4 is an SRG solution, not that all solutions that SGD finds are SRG. Regarding the justification of this claim, as the reviewer dZSf correctly pointed out, there is a typo in the label, the plotted matrix is in fact $\tilde{M}_5^T\tilde{M}_5.$ However, we note that the gram matrix looks exactly the same (up to scalar scaling) for all other layers (please see our global response where we upload a PDF showing all 5 layers). This can be also seen theoretically since all the involved matrices have by the construction of the SRG solution the same gram matrix. The reason why the gram matrices look exactly like this can be seen from the fact that the $H_l$ matrices are defined to be incidence matrices of the complete graph and their gram matrix is therefore naturally the adjacency matrix of the triangular graph. For a more detailed explanation, we refer you to Lemma 15 and Lemma 16, which are basically entirely devoted to compute the gram matrices and their entries and singular values explicitly. These calculations exactly agree with the plot in Figure 4. We will explain this in more detail in the revision.
>
> **Do numerical experiments include the ReLU activation?**
>
>  Yes, we include the ReLU activation in all our experiments in all layers.
>
> **What do we mean by “the rank deficiency is strongest in the middle layers of the MLP head” and how can one see this?**
>
> Thank you for this question, we will clarify the sentence. We do not mean that the rank in these middle layers is smaller than in the other layers, but rather that the tail singular values are closer to zero than the singular values of the other layers (i.e., 0.001 instead of 0.01). In fact, by looking closely at the green and red curves corresponding to layers 3 and 4, one can notice that their right tails are slightly below those of the other curves, meaning that the corresponding singular values are closer to 0 than those in the other layers.
>
> **Is the assumption on feature class-means being non-negative valid in experiments?**
>
> This is a great question. Indeed, in our experiments we observe that the intermediate features of the MLP head are non-negative before the application of the  ReLU, regardless of whether we obtain the SRG solution or not. We will clarify this point in the revision.

---

> ### Comment · Reviewer_hQr6 · 2024-08-09
>
> Thank you for addressing the concerns. I have increased my score based on the responses.
>
> One more suggestion to consider [optional]
> - I think it would be better to also quantitatively measure the rank. For example, a simple measure such as effective-rank can be used. If the notion of a low-rank used here is purely qualitative, and quantitative measures have limitations [if any], then such a discussion can further improve the paper quality.

---

> > ### Author Response · Authors · 2024-08-12
> >
> > Thank you for reconsidering your evaluation.
> >
> > Thank you for suggesting using effective ranks, it is an interesting idea. We will shortly report results with effective rank measures in the revision. However, to best convey our message the best rank measure is hard rank or its thresholded approximation.

---

> > > ### Author Response · Authors · 2024-08-13
> > > **Effective ranks for Figure 4**
> > >
> > > For completness of the discussion, we computed mean effective ranks (computed as exponential of the entropy of singular values after normalizing to sum one) for the two series of experiments in the Figure 4. Here are the results:
> > >
> > > CIFAR10: {'layer_1_er': 8.96, 'layer_2_er': 7.46, 'layer_3_er': 6.88, 'layer_4_er': 7.04, 'layer_5_er': 7.73, 'layer_5_er_post_relu': 9.52}
> > >
> > > MNIST: {'layer_1_er': 6.82, 'layer_2_er': 5.43, 'layer_3_er': 5.37, 'layer_4_er': 5.44, 'layer_5_er': 6.12, 'layer_5_er_post_relu': 9.47}
> > >
> > > Although the main message of our paper is orthogonal to to these quantitative values, they still reinforce our side claim that the low-rank bias is most pronounced somewhere in the middle of the MLP head, something we discussed previously. Thanks for suggesting this, there is indeed some value to these numbers, we will add them to the manuscript.

---

### Official Review · Reviewer_jQsy · 2024-07-12

**Soundness:** 4
**Presentation:** 4
**Contribution:** 3
**Rating:** 6
**Confidence:** 2

**Summary:**

Deep Neural Collapse (DNC) refers the the neural collapse phenomenon that has been observed on intermediate layers of a deep network (nb. whereas neural collapse (NC) focuses only on the penultimate layer). Similar to Unconstrained Feature Model (UFM) for analyzing NC, Deep UFM (DUFM) is the comparable framework to study DNC. However, as opposed to UFM for NC, it's been observed that once you go beyond two layers or two classes, DNC is no longer optimal for DUFM.

In this paper, the authors suggest that the reason for this is a low-rank bias of commonly used multi-layer regularization schemes, meaning the low-rank bias leads to optimal solutions that are lower rank than the neural collapse.

This paper extends the current research by looking at DUFM when there are more than two classes and the model is non-linear. Previous work has focused only on DUFM for binary classification with non-linear models or deep linear models or shallow models with multiple classes.

Most notably, they show that in this more complicated setting, DNC is not optimal wrt DUFM. Namely, the class means fail one of the original conditions of NC; i.e., they fail to form an orthogonal frame. The authors suggest that the reason for this is the low-rank bias under L2 regularization. It is worth noting that the condition DFC1 (which states within class variability goes to zero) is still optimal and it's only the DNC2 property (which states that class means are orthogonal) that conflicts with the low-rank bias.

Their findings are supported with strong theoretical proofs and empirically on benchmark datasets such as MNIST and CIFAR10.

One of the main takeaways from this paper is that…if a DNC solution is found, it is not because of DUFM or global optimality. Instead, it is because of an implicit bias of the optimization procedure.

**Strengths:**

The literature review is excellent and they do a great job motivating the problem they aim to address and how it compares with the existing literature.
The mathematical notation is clear and precise. The theory itself is very well developed with careful and detailed proofs.
The experiments they cover are solid and in agreement with their theory.

The authors pose some interesting open questions for future work related to the DNC1 condition.

**Weaknesses:**

Based on my understanding of the current work, I don't see any major weaknesses. I have some minor points of clarification or questions below but no major flaws.

**Questions:**

line 209 (clarification): Does the statement regarding K=2 or L=2 indicate that the argument of your theorem also be used to show DNC is optimal in this setting as shown in [51, 48] or are you just saying that as comparison when K=2 or L=2 by [51, 48] we know DNC is in fact optimal?

line 343: why the MSE loss? I didn't catch that part.

(typos/nits)
line 206 (typo): according to Theorem 5, should L \geq 3 be replaced with L = 3?

---

> ### Author Rebuttal · Authors · 2024-08-02
>
> Thank you for the comments and the positive evaluation of our paper. We address all questions below.
>
> **line 209 (clarification) about K=2, L=2**
>
> Our statement was only meant to say that the cases $K=2$ and $L=2$ were already treated in prior work [51, 48]. We will clarify this in the manuscript.
>
> **Why the MSE loss?**
>
> We focus on the MSE loss primarily for theoretical reasons,  because the MSE loss allows for a much more explicit analysis of the loss function. However, we expect that this is without much loss of generality, as both the MSE loss and the CE loss have similar behavior for well-fitting solutions. In addition, the low-rank bias does not come from the fit part of the loss, but rather from the regularization.
>
> **Typo:**
>
>  Thank you for pointing this out, we will correct it in the revision.

---

> > ### Comment · Reviewer_jQsy · 2024-08-10
> >
> > Ok. Thank you for the clarification. I have no further concerns. I'll keep my score as it is. Nice work.

---

> > > ### Author Response · Authors · 2024-08-12
> > >
> > > Thank you for your appreciation.

---

### Official Review · Reviewer_dZSf · 2024-07-12

**Soundness:** 3
**Presentation:** 3
**Contribution:** 3
**Rating:** 6
**Confidence:** 4

**Summary:**

This paper theoretically explores the deep neural collapse (DNC) phenomenon within non-linear deep models for multi-class classification. Neural collapse (NC) is a phenomenon in deep overparameterized networks where, the last layer's feature vectors align with class means and their corresponding classifier vectors, while also maximizing separation between classes. Previous works have studied the propagation of this symmetric geometry to earlier layers and proved its optimality which is termed as DNC. These works focus on DNC's optimality in multi-class linear networks or binary non-linear networks. This paper, however, extends the analysis to multi-class non-linear deep networks with ReLU activation, using the deep unconstrained features model (DUFM), where input features are treated as unconstrained parameters in the training objective.

While in the binary or linear case, DNC was shown to be optimal across all layers of DUFM, the authors show that in the multi-class case and in the presence of non-linearity, DNC is not always optimal. They prove this by explicitly constructing a solution, called SRG, that achieves a lower objective value than DNC. They show SRG is rank-deficit and the rank/loss gap between SRG and DNC increases as the network becomes deeper or the number of classes increases. They further numerically show on DUFM that, although DNC is not optimal, wider networks have an implicit bias toward finding the DNC solution. To further support their findings, they conduct additional experiments on benchmark datasets MNIST and CIFAR10 and ResNet architecture.

**Strengths:**

This paper closes a gap in the theoretical analysis of the NC literature. The result is interesting as it challenges the optimality of DNC established in previous works and the accuracy of the abstract UFM for modeling the dynamics of deep overparameterized models.

**Weaknesses:**

The proof of non-optimality of DNC in the paper relies on constructing a solution (SRG) that has a lower objective value. However, SRG might not be optimal. So, showing that SRG has a lower rank does not necessarily mean that the optimal solution also follows this low-rank structure. Still, the results and observations presented in this work remain interesting.

**Questions:**

1. As discussed in the paper, the last-layer feature vectors before the activation ($\tilde{M}_L$), have a lower rank, and after activation ($M_L$) recover the full rank structure of NC. This is also clear from Fig. 2 (right). However, what I find missing is whether after activation $M_L$ follows the NC geometry or not, i.e., $M_L \propto I_K$? Or does it follow a different rank $K$ structure?

2. Line 263 mentions that only for a few runs the training solutions match SRG. In the rest of the runs, do the solutions achieve a lower objective value than SRG (thus proving that SRG is a local optimal at best)?

3. What about $W_\ell$'s? Do they also exhibit a similar low-rank structure in the SRG solution as well as the numerical experiments?

    I also don't see why DNC3 is not well-defined in this setup (line 250). Still, at optimality, we might have the mean feature vectors $M_\ell$ and classifiers $W_\ell$ align in a low-rank structure. This alignment property doesn't hold in either the SRG or SGD solutions in the experiments?

4. Is there a small typo in Fig. 4 caption? The bottom right heatmap cannot be $M_5$ since the features after ReLU are non-negative and the heatmap cannot have negative values.

**Limitations:**

Yes

---

> ### Author Rebuttal · Authors · 2024-08-02
>
> Thank you for the comments and the positive evaluation of our paper. We address your concerns and questions below:
>
> **While the SRG solution has low-rank, it is not clear whether a globally optimal solution also has one.**
>
> This is a great point. We agree that we do not rigorously show this in our paper. However, we provide an intuitive explanation in lines 140-150. Our view is also aligned with a line of work on low rank bias [1, 2, 3]. In fact, [2] even proves the global optimality of low-rank solutions for sufficiently deep networks. Moreover, we highlight that the reason why SRG achieves a better loss than DNC is primarily its low-rank structure, as can be seen from our Lemma 13 which shows that the cost of intermediate layers linearly depends on their rank. Nevertheless, proving that global optima have low rank in our setting remains an interesting open problem.
>
> **Do the last layer’s post activation features follow the NC geometry? Or a different structure?**
>
> In our experiments, we do not recover the exact NC2 geometry, see Figure 2 (right) or Figure 4 (middle). In fact, the singular values are not all the same, and the tail is around 2-3x smaller than the largest singular value, which implies that the matrix cannot be orthogonal. This effect could be due to the large weight decay considered in our experimental setup. We also do not observe any specific type of different structure in the last layer.
>
> **Do the solutions that don’t achieve SRG structure outperform the SRG solution?**
>
> We did not find a solution that outperforms the SRG construction when the number of classes $K$ and number of layers $L$ are moderate. However, when $K$ and $L$ are very large, there are indeed solutions that outperform the SRG construction. This indeed shows that the SRG solution is not optimal in general, and we suspect that even lower rank structures are optimal in those settings.
>
> **Do the weight matrices also exhibit low-rank structure? What about the DNC3?**
>
> Yes, the weight matrices exhibit the same low-rank structure as features. This can be seen from the formula $W_l=H_{l+1}H_l^\dagger$ for the min-norm interpolator. This shows that the rank of $W_l$ is at most the minimum between the rank of its input and the rank of the output layer features. If those two are the same, $W_l$ must match the rank of the features exactly. Regarding DNC3, in our numerical experiments we do not observe exact alignment of rows of weight matrices with columns of feature matrices, and we do not expect this to happen theoretically. In fact, the rows of the weight matrices are not aligned with the columns of the feature matrices in our SRG solution.
>
> **Typo in Figure 4?**
>
>  You are right, thank you for spotting this. The layer in question is in fact $\tilde{M}_5$ instead of $M_5.$
>
> ----
>
> [1] G. Ongie and R. Willett. "The role of linear layers in nonlinear interpolating networks." arXiv preprint arXiv:2202.00856 (2022).
>
> [2] A. Jacot. "Implicit bias of large depth networks: a notion of rank for nonlinear functions." ICLR, 2023.
>
> [3] A. Jacot. "Bottleneck structure in learned features: Low-dimension vs regularity tradeoff." NeurIPS, 2023.

---

> > ### Comment · Reviewer_dZSf · 2024-08-11
> >
> > Thanks for your detailed response. I find the results interesting, and I believe clarifying these additional points can help with the message. I'll maintain my score.

---

> > > ### Author Response · Authors · 2024-08-12
> > >
> > > Thank you for your appreciation.

---

### Official Review · Reviewer_kT8c · 2024-07-13

**Soundness:** 4
**Presentation:** 4
**Contribution:** 4
**Rating:** 9
**Confidence:** 5

**Summary:**

Papyan et. al that showed that at the terminal phase of training there are four phenomena called neural collapse on the last layer of a deep neural network architecture. The authors extended Papyan's work by considering the earlier layers of nonlinear deep neural networks. They proved the first neural collapse condition (convergence to class means) is optimal for DNNs (Theorem 6 in manuscript). They also proved the second neural collapse condition (within-class variability) does not hold if number of layers and classes are high enough by showing there exists a scenario with better performance (Theorem 5). The claims are also supported by the experiments.

**Strengths:**

Authors extended our understanding of how deep neural networks work by analyzing the neural collapse for deep neural networks.

**Weaknesses:**

The results only hold under certain assumptions such as using multi-layer regularization.

**Questions:**

-

**Limitations:**

The authors adequately addressed the limitations.

---

> ### Author Rebuttal · Authors · 2024-08-02
>
> Thank you for your positive review. If any questions come up during the discussion phase, we’ll be happy to address them.

---

> > ### Comment · Area_Chair_Qoga · 2024-08-12
> > **Area Chair to Authors**
> >
> > Authors: Unfortunately this review will not be considered as part of the decision, unless the reviewer updates their review to include a justification of the score. The reviewer has already been reminded twice, but the review is unusable in its current form.

---

> > > ### Comment · Reviewer_kT8c · 2024-08-12
> > >
> > > Dear Area Chair,
> > >
> > > I kept my review brief as I think the contributions are clear and significant, and the manuscript is well-written. As you requested for further justification, I would like  to provide additional comments on why I believe this manuscript is very strong:
> > >
> > > i) The paper builds upon the foundational work by Papyan et al., who identified neural collapse at the terminal phase of training. By extending this analysis to earlier layers in nonlinear deep neural networks, the authors address an important gap in the literature. This extension provides a more comprehensive view of how neural collapse manifests across the entire network, not just in the final layer. Previous research following Papyan's work primarily focused on simple models such as linear networks and indicated that deep neural collapse is optimal across all layers. To my knowledge, it is the first work showing that neural collapse may not be optimal in certain layers of certain models.
> > >
> > > ii) The manuscript makes significant theoretical contributions. Specifically, Theorem 6 provides evidence that the convergence to class means in neural collapse is optimal for all layers. As most of the models used in practice have many layers, it extends the application of neural collapse significantly. Conversely, Theorem 5 challenges the validity of the within-class variability condition under certain conditions, suggesting that there are scenarios where performance could be optimized beyond the classical framework. These theorems provide new insights that could reshape existing theories and practices about how deep neural networks work.
> > >
> > > iii) The authors support their theoretical claims with many experimental results. The experiments not only validate the theoretical findings but also provide practical insights into how the neural collapse phenomena can be observed and leveraged in real-world scenarios.

---

> > > > ### Comment · Area_Chair_Qoga · 2024-08-12
> > > > **reply**
> > > >
> > > > Thank you. We can now incorporate this review.

---

### Author Rebuttal · Authors · 2024-08-02

We thank all the reviewers for their reviews. Here, as part of an answer to reviewer hQr6's question, we upload a PDF representing the gram matrices of class-means in all 5 layers of the SRG solution corresponding to the lower row (right) of Figure 4. As can be seen, all gram matrices (up to small imperfections) match the same SRG structure of the triangular graph.

---

### Decision · Program_Chairs · 2024-09-25

**Decision:**

Accept (poster)

**Comment:**

This paper studies the Neural Collapse phenomenon (NC) in deep nonlinear networks. They prove that a collapsed solution is not optimal under the widely-used “deep unconstrained feature model.” This highlights a surprising aspect of neural collapse in deep nonlinear networks, which was not present in prior analyses. Further, the authors explore the low-rank bias of regularization as a potential mechanisms for this suboptimal behavior.

Reviewers agree that this is a strong, well-motivated technical contribution which yields insight on a timely topic. I recommend acceptance.